# Continual Memorization of Factoids in Language Models

**Howard Chen**[*]                                                    *howardchen@cs.princeton.edu*

**Jiayi Geng**[*]                                                          *jiayig@princeton.edu*

**Adithya Bhaskar**                                                   *adithyab@princeton.edu*

**Dan Friedman**                                                *dfriedman@cs.princeton.edu*

**Danqi Chen**                                                        *danqic@cs.princeton.edu*

*Princeton Language and Intelligence (PLI), Princeton University*

**Reviewed on OpenReview:** *https://openreview.net/forum?id=5Yd5QAKIFR*

## Abstract

As new knowledge rapidly accumulates, language models (LMs) with pretrained knowledge quickly become obsolete. A common approach to updating LMs is fine-tuning them directly on new knowledge. However, recent studies have shown that fine-tuning for memorization may be ineffective in storing knowledge or may even exacerbate hallucination—raising doubts about its reliability when applied repeatedly. To study this, we formalize the problem of *continual memorization*, where a model must memorize and retain a set of factoids through multiple stages of fine-tuning on subsequent datasets. We first characterize the forgetting patterns through extensive experiments and show that LMs widely suffer from forgetting, especially when needing to memorize factoids in the second stage. We posit that forgetting stems from suboptimal training dynamics which fail to: (1) protect the memorization process when learning factoids or (2) reduce interference from subsequent training stages. To test this hypothesis, we explore various data mixing strategies to alter the fine-tuning dynamics. Intriguingly, we find that mixing randomly generated word sequences or generic data sampled from pretraining corpora at different training stages effectively mitigates forgetting (REMIX: Random and Generic Data Mixing). REMIX can recover performance from severe forgetting, outperforming replay methods and other continual learning baselines. We analyze how data mixing can influence the learning process and find that robust memorization follows a distinct pattern—the model stores factoids in earlier layers than usual and diversifies the layers that retain them, which results in easier recall and manipulation of the learned factoids.[1]

## 1 Introduction

Language models (LMs) have shown a remarkable ability to absorb massive amounts of knowledge through large-scale pretraining (Petroni et al., 2019; AlKhamissi et al., 2022; Cohen et al., 2023). However, knowledge is not static—new information accumulates quickly while old knowledge becomes obsolete. This dynamic nature necessitates frequent model updates, making costly pretraining impractical. A common approach is to fine-tune the model directly on new knowledge. However, recent studies have shown that fine-tuning is brittle: training on long-tail knowledge can lead to unintended disruptions, such as decreased factuality and exacerbated hallucinations (Kang et al., 2025; Gekhman et al., 2024; Zhang et al., 2024). Furthermore, a

---

[*]Equal contribution.
[1]Code and data are available at `https://github.com/princeton-nlp/continual-factoid-memorization`.

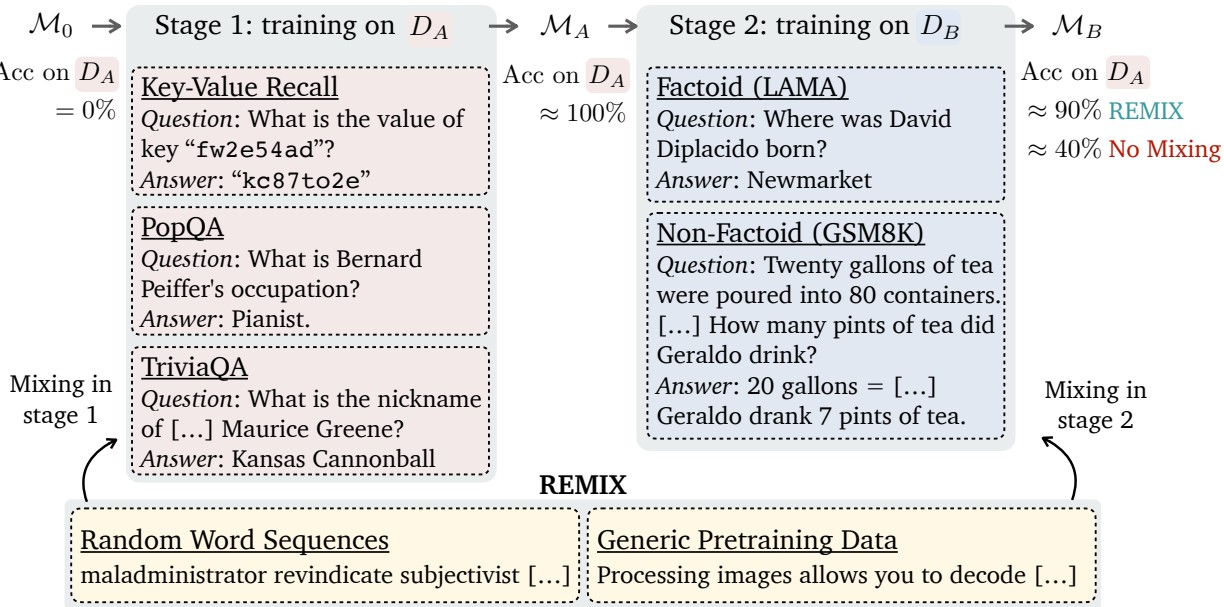

Figure 1: The continual memorization setting. In stage 1 (red box), a pretrained model $\mathcal{M}_0$ is trained to convergence on a factoid dataset $D_A$ to obtain model $\mathcal{M}_A$. In stage 2, model $\mathcal{M}_A$ is further trained on either a factoid dataset or a non-factoid dataset (blue box) to obtain model $\mathcal{M}_B$. The final model $\mathcal{M}_B$ is evaluated on the training examples $D_A$ in stage 1. REMIX: mixing random words and pretraining data into training during stages 1 and 2 alleviates forgetting.

fine-tuned model might not properly memorize knowledge, failing to recall or manipulate it effectively (Allen-Zhu & Li, 2024; 2025). This fragility raises the question of whether fine-tuning can be applied repeatedly as a reliable mechanism for continual knowledge acquisition.

To address this, we investigate the dynamics of memorization in a continual learning setting (McCloskey & Cohen, 1989; Ratcliff, 1990), in which the model acquires new information incrementally, one set at a time. While prior research on continual learning in LMs has focused on general capabilities such as reasoning (Luo et al., 2023a) or broad proxies like language modeling loss over a general corpus (Yildiz et al., 2024), we use this framework to study the memorization dynamics of LMs through fine-tuning. We formalize this as *continual memorization*, in which a model is first trained on a small collection of factoids (factual associations) and must retain this knowledge after training on additional datasets in a subsequent stage. Specifically, we train models to memorize factoid datasets (stage 1) and then evaluate how well these factoids are retained after a second stage of training on a different dataset (Figure 1). We study two stages in our main setting, and provide analysis of more stages in §5.

We conduct extensive experiments to characterize forgetting patterns in continual memorization. First, we find that the most severe forgetting occurs when the second stage of training involves another factoid dataset, regardless of whether the facts overlap with those from stage 1. For example, accuracy on TriviaQA drops from 100% (after stage 1) to 39.8% after further training on other factoid datasets, such as LAMA. Forgetting is less pronounced when fine-tuning on non-factoid datasets, such as those involving coding, math, or chat. We also find that long-tail data is the hardest to retain—a randomly generated key-value string is most susceptible to forgetting, with accuracy dropping to 13% after further training on another factoid dataset. This aligns with recent findings that memorizing knowledge (e.g., factoids) causes greater disruption to other model capabilities, which, in our setting, results in more severe forgetting (Kang et al., 2025). Finally, we observe that common experience replay methods, which mix a fraction of data from earlier training stages, fail to prevent forgetting when the second stage involves a factoid dataset, in contrast to their effectiveness in general continual learning settings. For instance, even when mixing in 10% of the factoids from stage 1, the model fails to recover performance beyond 60%.

Next, we investigate whether data mixing strategies can mitigate forgetting. Through theoretical derivations, we develop intuition that this question may be approached in two ways: 1) teaching the model to *protect learned knowledge better* in the first stage, or 2) *reducing the interference* of the second stage by manipulating the data distribution. Based on this hypothesis, we examine a range of data mixing strategies at each stage. Intriguingly, we find that mixing in either *generic pretraining data* or even *random word sequences* leads to a considerable reduction in forgetting. We combine both strategies, and refer to this mitigation as REMIX (Random and Generic Data Mixing). Our experiments demonstrate that REMIX is highly effective at helping the model retain learned factoids: in the most severe case, REMIX increases post-stage 2 accuracy from 13.5% to 53.2%. In comparison, replay can only reach 41.6% despite using 10% of the factoids from stage 1. Other common continual learning methods also fall short: weight regularization (EWC) Kirkpatrick et al. (2017) and behavior regularization Sun et al. (2020) both lag behind REMIX. These benefits are seen consistently across several choices of factoid and non-factoid tasks in stage 2.

To understand why these mixing strategies help reduce forgetting, we analyze REMIX using Logit Lens (nostalgebraist, 2020) and ablation studies. Our analysis suggests that including a broad range of mixed data encourages the model to store facts in relatively earlier layers (compared to the baseline setting), as well as to diversify where it stores the knowledge. This diversification allows it to better protect learned knowledge in subsequent stages of training. In the second stage, jointly learning the mixing data and the stage 2 data helps prevent overfitting to a narrow distribution, alleviating the negative interference on the learned factoids. Finally, we show that more robustly memorized factoids are not only better retained and recalled, but are more easily extracted for manipulation.

We summarize our contributions as follows:

- We formalize the *continual memorization* setting and demonstrate the fragility of the factoid memorization process in LMs; we further show that it cannot be easily addressed with replay.

- We find that mixing random and generic data (REMIX) in different stages can greatly mitigate forgetting without accessing the factoids from prior stages.

- We find that successful mixing diversifies the layers where the learned knowledge is stored and tends to store it in earlier layers than in models that suffer from forgetting, shedding light on the patterns of robust memorization.

## 2 Continual Memorization of Factoids

### 2.1 Problem Definition

**Factoid vs non-factoid datasets.** We define a *factoid* to be a triplet (subject, relation, object). A dataset $D \in \mathcal{D}$ in this paper is a set of (prompt, response) pairs. A *factoid dataset* $D \in \mathcal{D}_{\text{fact}} \subset \mathcal{D}$ is a set of factoids formatted as pairs (e.g., "prompt = The `<relation>` of `<subject>` is" → response = `<object>`). If $D \in \mathcal{D} \setminus \mathcal{D}_{\text{fact}}$, we call $D$ a *non-factoid* dataset. A language model $\mathcal{M}$: $p_\theta(y \mid x)$ parameterized by $\theta$ defines a distribution over response $y$ given the prompt $x$. Given a model $\mathcal{M}$ and dataset $D$, we denote by $\mathcal{L}(\theta; D) \in \mathbb{R}^+$ the loss and $\mathcal{A}(\mathcal{M}; D) \in [0,1]$ the average exact-match accuracy. We define a factoid $x$ to be *familiar* to $\mathcal{M}$ if $\mathcal{A}(\mathcal{M}; \{x\}) = 1$ and *unfamiliar* otherwise. [2] An unfamiliar dataset consists entirely of unfamiliar facts.

**Continual memorization.** We now describe the setting of *continual memorization*, which consists of two or more stages. We describe the setting with two stages below. Let $D_A \in \mathcal{D}_{\text{fact}}$ be a factoid dataset, and $D_B \in \mathcal{D}$ be another dataset (factoid or non-factoid). In the first stage, a pretrained model $\mathcal{M}_0$ is trained on $D_A$ until convergence to obtain the trained model $\mathcal{M}_A$ with near-zero loss $\mathcal{L}(\theta_A; D_A) \approx 0$ and accuracy $\mathcal{A}(\mathcal{M}_A; D_A) \approx 1$. In the second stage, $\mathcal{M}_A$ is further trained on $D_B$ until convergence. The resulting model $\mathcal{M}_B$ is evaluated on $D_A$ to gauge its retention $\mathcal{A}(\mathcal{M}_B, D_A)$. In this paper, we consider the case where all factoid datasets (in the first as well as second stage—if applicable) are unfamiliar and we refer to

---

[2]Drawing on Kang et al. (2025) and Gekhman et al. (2024), we distinguish between familiar and unfamiliar factoids, as training on unfamiliar instances can disrupt model behavior in ways that make forgetting patterns more apparent.

them simply as factoid datasets. Typically, one observes $\mathcal{A}(\mathcal{M}_B, D_A) \ll \mathcal{A}(\mathcal{M}_A, D_A)$ due to catastrophic forgetting. Figure 1 illustrates this setting.

## 2.2 Constructing Factoid Datasets

We consider a variety of (unfamiliar) factoid datasets in our experiments. These datasets are either 1) constructed synthetically to ensure that they were not seen by the model $\mathcal{M}_0$ during pretraining—such as by generating random key-value mappings, or 2) filtered from factoid datasets to remove familiar instances (details in § B.8). We further describe the specific choice of datasets for the two stages below.

**Stage 1: Factoid dataset $D_A$.** Key-Value Recall (KVR): we generate $2,000$ unique key-value pairs, each containing 8 characters from the mix of alphabets and number digits. PopQA: we sample $2,000$ unfamiliar knowledge triplets from a set of diverse questions and relationships about long-tail entities (Mallen et al., 2023). TriviaQA: we sample $2,000$ unfamiliar question-answer pairs from the dataset (Joshi et al., 2017). See examples in Figure 1 and §B.9.

**Stage 2: Dataset $D_B$.** We explore a wide range of datasets in stage 2 to reflect real-world application scenarios. Specifically, we consider two types of datasets: factoid and non-factoid. We choose this split because we want to see how the effect of stage 2 changes from a knowledge-intensive factoid dataset to, e.g., a general instruction tuning dataset. Additionally, domain-specific knowledge and instruction-tuning data represent two of the most common types of data used for supervised fine-tuning—a fact reflected in our selection of tasks. We explore:

- Factoid datasets: LAMA (Petroni et al., 2019), Entity Questions (Sciavolino et al., 2021), WebQA (Berant et al., 2013). In addition, we also explore adding new (and unfamiliar) examples from the distribution of $D_A$ (i.e., the same task as in stage 1) referred to as the "In-Domain" (ID) datasets in our results.

- Non-factoid datasets: UltraChat (Ding et al., 2023), EvolCode (Luo et al., 2023b), APPS (Hendrycks et al.), GSM8K (Cobbe et al., 2021), and MATH (Hendrycks et al., 2021b). These datasets exemplify common non-factoid datasets used for finetuning: chat, code and math.

**Training and evaluation.** We use Llama-3-8B (Dubey et al., 2024) and Mistral-7B (Jiang et al., 2023) to initialize $\mathcal{M}_0$ in our experiments (both are base models). All of our experiments use the Tulu-v2 prompt template (Ivison et al., 2023), i.e., `"<user>...<assistant>..."` for both stages. We provide training details in §B.8. Our accuracies are computed as Exact String Match and normalized to $[0, 100]$ for all the experiments, as the tasks only need to generate a few tokens. We report averaged accuracy across 3 runs.

## 3 How Do Models Forget Factoids?

### 3.1 Understanding the Forgetting Patterns

We first establish the forgetting patterns in continual memorization by examining which intervening tasks affect the final accuracy most severely when used in the second stage. Table 1 shows the performance degradation of stage 1 tasks after training on stage 2 tasks. We observe that forgetting is most severe when stage 2 is also a factoid dataset, degrading accuracy for KVR to 13.5%, PopQA to 47.0%, and TriviaQA to 39.8% on average. In fact, with LAMA these accuracies fall to 2.1%, 7.7% and 4.3% respectively—far below the numbers seen with non-factoid datasets. This corroborates findings from the continual learning literature which suggest catastrophic forgetting happens when two tasks are similar and therefore cause interference (Farajtabar et al., 2020; Bennani et al., 2020; Doan et al., 2021). While the overall trend holds, we note caveats in §B.11.

### 3.2 Replay Does Not Mitigate Forgetting Fully

| $D_A$ | $D_B$: **Factoid** | | | | | $D_B$: **Non-Factoid** | | | | | |
|---|---|---|---|---|---|---|---|---|---|---|---|
| | ID | LAMA | EQ | WQ | **Avg** | GSM8K | MATH | EC | APPS | UC | **Avg** |
| KVR | 0.5 | 2.1 | 17.4 | 33.8 | 13.5 | 24.4 | 27.3 | 49.5 | 26.7 | 66.6 | 38.9 |
| PopQA | 49.8 | 7.7 | 57.8 | 72.5 | 47.0 | 19.0 | 92.4 | 77.0 | 55.1 | 48.5 | 58.4 |
| TriviaQA | 45.6 | 4.3 | 40.5 | 68.6 | 39.8 | 9.4 | 87.6 | 54.4 | 70.4 | 67.6 | 57.9 |

Table 1: Forgetting in continual memorization. Lower accuracies imply more forgetting. All stage 1 datasets are trained to 100% accuracy before stage 2 training. The lowest accuracy in each row is underlined, and "ID" signifies that we use unseen examples from $D_A$ to form the dataset in the second stage ($D_B$). We see that factoid datasets cause greater forgetting than non-factoid datasets when used in stage 2. (EQ = EntityQA, WQ = WebQA, EC = EvolCode, and UC = UltraChat.)

Replay-based methods mitigate forgetting by sampling a small portion of data from earlier stages and mixing it with the subsequent dataset during training. Replay from past experience has been a long-established mitigation to prevent forgetting in reinforcement learning research (e.g., Mnih et al., 2013) and more recently continual pretraining for LMs. Although replay-based methods have proven helpful for continual learning, we hypothesize that they will be less effective for tasks requiring memorization, as the individual instances are largely unrelated (Feldman, 2020; Yang et al., 2023). Figure 2 shows that although replay reduces forgetting across the board, the effectiveness is not uniform. Replay is less effective at preventing forgetting when stage 2 is factoid than non-factoid (full

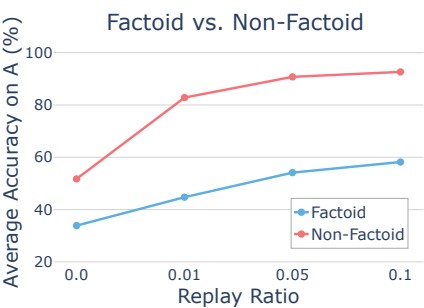

Figure 2: Replay results averaged across all $D_B$ for four mixing ratios.

results in §B.4). The experiments suggest that manipulating the training dynamics such as exposing the model to different distributions can affect the model's ability to recall factoids, even when the replayed factoids are individually independent from other factoids in the same stage.

## 4 REMIX: Random and Generic Data Mixing

### 4.1 Method

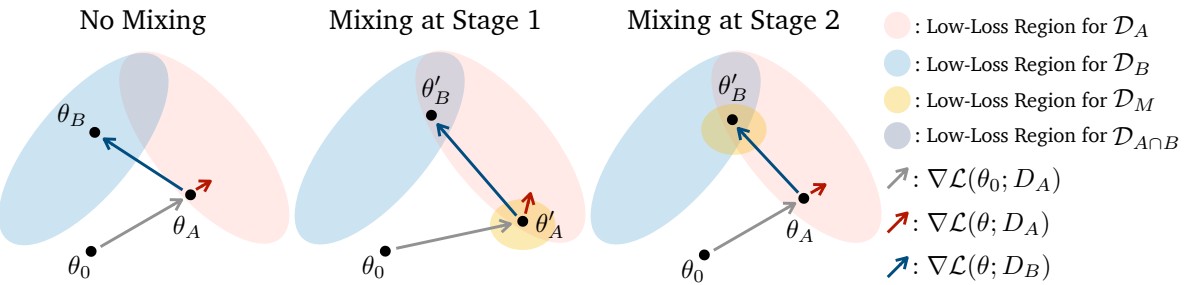

Figure 3: Intuition behind each mixing strategy. In general, forgetting occurs when $\nabla\mathcal{L}(\theta; D_A)^T \nabla\mathcal{L}(\theta; D_B) < 0$ (angle between red and blue arrows larger than 90 degree). The model goes from $\theta_0$ to $\theta_A$ in stage 1 (gray arrow), and arrives at $\theta_B$ in stage 2 (blue arrow). The translucent blobs represent low-loss region for each dataset. **No Mixing**: the opposing angle between the red and blue arrows contributes to forgetting. **Mixing at Stage 1**: the mixing data $D_M$ protects memorization by shifting the model parameters to reduce the angle between the red and blue arrows while converging to a low loss on $D_A$. **Mixing at Stage 2**: mixing data $D_M$ reduces the interference of $D_B$ by lowering the angle between blue and red arrows.

Despite the shortcomings of replay, we make one key observation: when mixing only 10% of the factoids used in stage 1, the accuracy increases after learning non-factoid data in stage 2 from no mixing at 40.1% to 83.9%

(Table 9). This implies the existence of associations that were stored in model weights but could not be retrieved effectively. It is then prudent to ask if these "hidden" associations can be surfaced with a different choice of mixing data. To answer this question, we propose $\underline{R}$andom and $\underline{Ge}$neric Data $\underline{Mix}$ing (REMIX), a data mixing strategy that manipulates the memorization dynamics during training to prevent forgetting. The mixing data is sampled from either random word sequences or generic text such as pretraining corpora, which has no overlap with the factoids aiming to memorize in stage 1. Figure 3 illustrates the intuition behind the mixing strategies.

For the purpose of developing intuition, we take the simplification to assume the entire optimization is captured by the one-step gradient update. Let $\mathcal{L}(\theta; D)$ be the empirical loss on dataset $D$, and let $\nabla\mathcal{L}(\theta; D)$ denote its gradient. Starting from $\theta_0$, stage-1 training on $D_A$ yields

$$\theta_A = \theta_0 - \eta\nabla\mathcal{L}(\theta_0; D_A),$$

and subsequent stage-2 training on $D_B$ yields

$$\theta_B = \theta_A - \eta\nabla\mathcal{L}(\theta_A; D_B),$$

for learning rate $\eta > 0$. A first-order Taylor expansion of $\mathcal{L}(\cdot; D_A)$ around $\theta_A$ gives

$$\mathcal{L}(\theta_B; D_A) - \mathcal{L}(\theta_A; D_A) \approx -\eta\,\nabla\mathcal{L}(\theta_A; D_B)^T\,\nabla\mathcal{L}(\theta_A; D_A),$$

and catastrophic forgetting on $D_A$ occurs when $\nabla\mathcal{L}(\theta_A; D_B)^T\,\nabla\mathcal{L}(\theta_A; D_A) < 0$, i.e., when the gradients induced by $D_B$ point in a direction that increases the loss on $D_A$. REMIX changes the learning dynamics by mixing $D_M$ in the first stage and/or $D'_M$ in the second stage to prevent forgetting. We state the condition for REMIX to mitigate forgetting in the following.

**Proposition 1** (Forgetting mitigation condition for REMIX)**.** *Consider modified updates with mixing datasets $D_M$ (stage 1) and $D'_M$ (stage 2):*

$$\theta'_A = \theta_0 - \eta\nabla\mathcal{L}(\theta_0; D_A \cup D_M), \quad \theta'_B = \theta'_A - \eta\nabla\mathcal{L}(\theta'_A; D_B \cup D'_M).$$

*Assume $\mathcal{L}(\theta; D_A)$ is differentiable and locally well-approximated by its first-order expansion around $\theta_A$ and $\theta'_A$. We say REMIX is effective on $D_A$ if*

$$\mathcal{L}(\theta'_B; D_A) < \mathcal{L}(\theta_B; D_A).$$

*To first order, a sufficient condition for REMIX to be effective is*

$$\nabla\mathcal{L}(\theta'_A; D_B \cup D'_M)^T\,\nabla\mathcal{L}(\theta_A; D_A) \;\geq\; \nabla\mathcal{L}(\theta_A; D_B)^T\,\nabla\mathcal{L}(\theta_A; D_A).$$

*Equivalently, REMIX is beneficial if (i) mixing at stage 1 moves the model to a region where the gradients for $D_A$ and $D_B$ are less opposed, and/or (ii) mixing at stage 2 rotates the effective stage-2 gradient toward alignment with $\nabla\mathcal{L}(\theta_A; D_A)$, thereby reducing interference with previously memorized factoids.*

At stage 1, the mixing data can teach the model to diversify where to store the knowledge, resulting in a better starting position in the parameter space for stage 2 training (smaller angle between $\nabla\mathcal{L}(\theta; D_A)$ and $\nabla\mathcal{L}(\theta; D_B)$), achieving better *protection* of the memorized factoids. At stage 2, the mixing data can rotate the direction of $\nabla\mathcal{L}(\theta; D_B)$ to align with $\nabla\mathcal{L}(\theta; D_A)$, thus *reducing the interference* on the memorized factoids from stage 2 training; if the two gradients are in extreme opposing directions, it becomes easier for the mixing data to align them. We provide detailed derivations to concretize the intuition in §A.3. Based on the above insight, we posit: 1) *mixing at stage 1 mitigates forgetting most when the mixing data is unrelated to both $D_A$ and $D_B$*, and 2) *mixing at stage 2 is most effective if the forgetting is severe, and is more effective when $D'_M$ aligns with $D_A$.*

**REMIX datasets $D_M$.** We explore three data sources for generic data mixing: 1) Knowledge Pile (Fei et al., 2024), 2) Arxiv Pile (Gao et al., 2020), and 3) Fineweb (Penedo et al., 2024). We construct the Random Word Sequence data by collecting a set of uniformly sampled 50 random word sequences from the

NLTK Word Corpus (Bird et al., 2009). We check and ensure no overlap between the factoid data and the mixing data (see details in §B.10). When applying REMIX, we add the mixing data directly to $D_A$ in stage 1 and $D_B$ in stage 2. Therefore, the model trains on more data at each stage with mixing. We use Random Word Sequence and Knowledge Pile as the main datasets in the following experiments and later show that other mixing datasets show similar trends. We use $D_A : D_M = 1 : 2$ and $D_B : D_M = 1 : 2$ for the main experiments.

| | Factoid | | | | | Non-Factoid | | | | | |
| --- | --- | --- | --- | --- | --- | --- | --- | --- | --- | --- | --- |
| | ID | LAMA | EQ | WQ | **Avg** | GSM8K | MATH | EC | APPS | UC | **Avg** |
| **Key-Value Recall** | | | | | | | | | | | |
| No Mixing | 0.5 | 2.1 | 17.4 | 33.8 | 13.5 | 24.4 | 27.3 | 49.5 | 26.7 | 66.6 | 38.9 |
| Random / - | 8.9 | 2.5 | 42.5 | 61.4 | 28.8 | **64.1** | **75.9** | **85.3** | **75.0** | **89.1** | **77.9** |
| K-Pile / - | 0.1 | 0.0 | 3.2 | 30.1 | 8.4 | 47.3 | 58.4 | 62.2 | 19.0 | 74.3 | 52.2 |
| - / Random | 0.2 | 0.1 | 2.9 | 5.3 | 2.1 | 15.1 | 11.7 | 33.8 | 16.5 | 66.8 | 28.8 |
| - / K-Pile | 0.8 | 40.0 | 36.4 | 33.9 | 27.8 | 12.8 | 8.8 | 40.5 | 16.8 | 70.2 | 29.8 |
| Random / K-Pile | **10.6** | **62.4** | **69.5** | **70.2** | **53.2** | 45.8 | 45.4 | 74.7 | 51.2 | 86.8 | 60.8 |
| **PopQA** | | | | | | | | | | | |
| No Mixing | 49.8 | 7.7 | 57.8 | 72.5 | 47.0 | 19.0 | 92.4 | 77.0 | 55.1 | 48.5 | 58.4 |
| Random / - | 62.0 | 17.7 | 69.3 | 65.8 | 53.7 | **51.4** | 89.3 | 82.7 | 81.8 | 66.0 | 72.2 |
| K-Pile / - | 24.0 | 2.8 | 11.3 | 31.8 | 17.5 | 46.4 | 92.7 | **94.0** | **87.2** | **90.9** | **82.2** |
| - / Random | 35.7 | 5.2 | 38.1 | 45.9 | 31.2 | 16.8 | 93.5 | 87.5 | 59.3 | 70.7 | 65.6 |
| - / K-Pile | **86.6** | **90.8** | **93.9** | 74.4 | **86.4** | 25.9 | **94.0** | 92.4 | 73.9 | 74.7 | 72.2 |
| Random / K-Pile | 82.6 | 85.8 | 90.7 | **80.5** | 84.9 | 38.5 | 88.7 | 88.3 | 79.2 | 74.4 | 73.8 |
| **TriviaQA** | | | | | | | | | | | |
| No Mixing | 45.6 | 4.3 | 40.5 | 68.6 | 39.8 | 9.4 | 87.6 | 54.4 | 70.4 | 67.6 | 57.9 |
| Random / - | 64.9 | 8.1 | 60.0 | 70.8 | 51.0 | 27.1 | **84.9** | 71.2 | 87.3 | 70.8 | 68.3 |
| K-Pile / - | 9.4 | 0.9 | 3.8 | 21.0 | 8.8 | **31.9** | 82.9 | **93.5** | **90.7** | **90.1** | **77.8** |
| - / Random | 25.0 | 5.5 | 19.9 | 38.8 | 22.3 | 4.1 | 81.0 | 84.0 | 62.2 | 71.6 | 60.6 |
| - / K-Pile | **90.8** | **90.1** | **91.5** | **89.8** | **90.6** | 2.8 | 79.1 | 75.9 | 53.7 | 69.8 | 56.3 |
| Random / K-Pile | 90.2 | 89.2 | 89.6 | 86.5 | 88.9 | 12.5 | 81.8 | 71.2 | 74.6 | 70.0 | 62.0 |

Table 2: REMIX results for Llama-3-8B with the combinations of $D_A$, $D_B$, and $D_M$. No Mixing denotes the original two-stage training without applying REMIX. Each $D_{M_1}$ / $D_{M_2}$ row represents mixing with $D_{M_1}$ in stage 1 and mixing with $D_{M_2}$ in stage 2. "-" indicates no mixing at that stage. All numbers are in accuracy and averaged across three runs. (EQ = EntityQA, WQ = WebQA, EC = EvolCode, and UC = UltraChat.)

## 4.2 Results

**Factoid tasks.** Table 2 shows the results of factoid tasks with Llama-3-8B. We observe that mixing Random Word Sequences prevents forgetting across the board, improving average accuracy for all $D_A$, improving Key-Value Recall (13.5% → 28.8%), PopQA (47.0% → 53.7%), and TriviaQA (39.8% → 51.0%). On the other hand, mixing Knowledge Pile at stage 1 hurts the performance. Mixing at stage 2 shows an opposite trend. We observe drastically better performance with mixing Knowledge Pile, improving the average accuracy for Key-Value Recall (13.5% → 27.8%), PopQA (47.0% → 86.4%), and TriviaQA (39.8% → 90.6%). In contrast, mixing Random Word Sequence at stage 2 exacerbates forgetting. The results align with our prediction that stage 1 mixing relies on data that is unrelated to either $D_A$ or $D_B$, while stage 2 mixing benefits most when forgetting is severe and the mixing data aligns with $D_A$.

**Non-factoid tasks.** Figure 2 shows that the model exhibits consistent results after training on non-factoid data at stage 2. We observe that stage 1 mixing is more beneficial than stage 2 mixing across the board. However, the best mixing data varies for different $D_A$. KVR benefits most from mixing Random Word Sequence at stage 1 (38.9% → 77.9%), while Knowledge Pile benefits most on PopQA (58.4% → 82.2%) and TriviaQA (57.9% → 77.8%).

**Applying mixing at both stages.** Based on the observation that mixing with Random Word Sequence at stage 1 and mixing Knowledge Pile at stage 2 individually benefit memorization intensive tasks, we examine

if the two stages can be combined. Figure 2 shows that the combination outperforms individual stage mixing, demonstrating the possibility of composing mixing strategies. We also provide stage 2 task performance in §B.2.

**Mistral results.**    We report REMIX results for Mistral in Table 3 (full results in §B.3). For KVR, REMIX can successfully prevent forgetting and improve performance after stage 2 training on factoid data ($15.0\% \rightarrow 43.5\%$). REMIX's advantages are less pronounced since the No Mixing baselines are not affected by forgetting too severely.

| | | LAMA | EntityQA | WebQA | **Avg** |
|---|---|---|---|---|---|
| **KVR** | No Mixing | 0.1 | 15.4 | 29.6 | 15.0 |
| | Random / K-Pile | 47.5 | 44.1 | 39.0 | 43.5 |
| **PopQA** | No Mixing | 66.9 | 92.3 | 89.6 | 82.9 |
| | Random / K-Pile | 90.5 | 92.3 | 89.0 | 90.6 |
| **TriviaQA** | No Mixing | 71.6 | 86.4 | 91.5 | 83.2 |
| | Random / K-Pile | 77.0 | 81.5 | 83.1 | 80.5 |

Table 3: REMIX results for Mistral-7B-v0.3 on Factoid benchmarks. We compare the No Mixing baseline to REMIX that mixes with Random Word Sequence at stage 1 and mixes with Knowledge Pile at stage 2. (EQ = EntityQA, WQ = WebQA.) We provide the complete results in §B.3

| | LAMA | EntityQA | WebQA | **Avg** |
|---|---|---|---|---|
| **KVR** | | | | |
| No Mixing | 2.1 | 17.4 | 33.8 | 17.8 |
| REMIX (Random / K-Pile) | 62.4 | 69.5 | 70.2 | **67.4** |
| Weight Regularization | 0.1 | 4.3 | 76.7 | 27.3 |
| Behavior Regularization | 0.2 | 15.6 | 36.6 | 17.5 |
| Parameter Expansion | 52.3 | 52.2 | 68.8 | 57.8 |
| **PopQA** | | | | |
| No Mixing | 7.7 | 57.8 | 72.5 | 46.0 |
| REMIX (Random / K-Pile) | 85.8 | 90.7 | 80.5 | **85.7** |
| Weight Regularization | 12.1 | 67.4 | 76.7 | 52.1 |
| Behavior Regularization | 7.5 | 59.3 | 55.5 | 40.7 |
| Parameter Expansion | 83.0 | 84.3 | 80.0 | 82.4 |
| **TriviaQA** | | | | |
| No Mixing | 4.3 | 40.5 | 68.6 | 37.8 |
| REMIX (Random / K-Pile) | 89.2 | 89.6 | 86.5 | **88.4** |
| Weight Regularization | 7.9 | 58.5 | 80.3 | 48.9 |
| Behavior Regularization | 6.8 | 39.0 | 71.0 | 38.9 |
| Parameter Expansion | 80.7 | 86.6 | 83.0 | 83.4 |

Table 4: Comparison of REMIX to the weight regularization, behavior regularization, and parameter expansion baselines with the factoid datasets at stage 2.

**Comparison to other baselines.**    We compare with three other representative baselines against REMIX in Table 4: 1) weight regularization Kirkpatrick et al. (2017), 2) behavior regularization Sun et al. (2020), and 3) parameter expansion (von Oswald et al., 2020). We use Elastic Weight Consolidation (EWC) for weight regularization and calculate the Fisher score using one backward pass using the current mini-batch for training. For behavior regularization, we add the KL between the training model vs the original reference model to the loss. We provide the parameter expansion based baseline using LoRA adaptors (Hu et al., 2022). We randomly select parameters to train and do not explicitly avoid overlapping of the stage 1 and stage 2 tunable parameters. We observe that the weight regularization baseline and behavior regularization baselines lag behind REMIX by a large margin (40% on KVR, 30% on PopQA, and 40% on TriviaQA).

# 5 Analysis

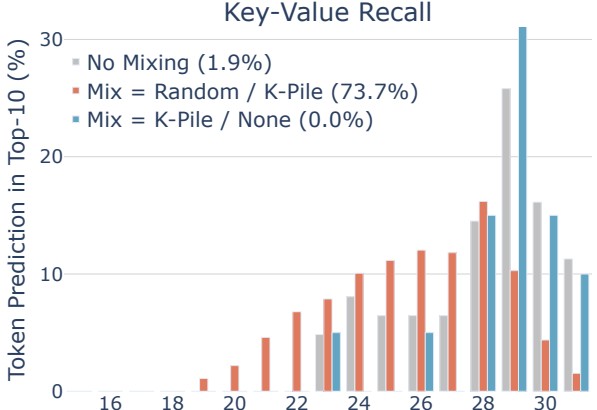

| | LoF Mean | LoF STD | Acc |
|---|---|---|---|
| **KVR** | | | |
| No Mixing | 26.8 | 15.9 | 40.6 |
| REMIX (R/K-Pile) | 25.9 | 21.6 | **80.0** |
| **PopQA** | | | |
| No Mixing | 23.5 | 5.1 | 73.0 |
| REMIX (R/K-Pile) | 22.1 | 5.0 | **91.0** |
| **TriviaQA** | | | |
| No Mixing | 24.2 | 4.7 | 68.0 |
| REMIX (R/K-Pile) | 22.6 | 5.0 | **92.0** |

Figure 4: Left: probing on Key-Value Recall using Logit Lens. x-axis: layer index. y-axis: the normalized frequency of the correct token occurring in the top-10 tokens probed at each layer. % following each legend shows the accuracy on each stage 1 task. Right: layer of first occurence (LoF) aggregated over 100 examples. The mean, standard deviation and overall accuracy on KVR, PopQA and TriviaQA. Lower mean in LoF and higher STD correlates with better performance.

**Robust memorization learns factoids in earlier layers.** We use Logit Lens (nostalgebraist, 2020) to decode the top 10 tokens from the representations at each layer using the output embedding. We record the layer index of the first occurrence of the correct token, referred to as layer of first occurrence (LoF). LoF is then normalized by the total number of occurrences. This measure indicates how early the correct token first appears. In Figure 4 (left), we compare 1) No Mixing, 2) Random / K-Pile which successfully prevents forgetting, and 3) K-Pile / None which suffers from forgetting for KVR. We notice two main differences between the two runs – first, the successful run moves the knowledge to an earlier layer, whereas the unsuccessful one does not change where the factoids are stored. The successful run also *diversifies* the set of layers that are used. We aggregate the mean and standard deviation of LoF over 100 examples in Figure 4 (right). The results corroborate our intuition: the model protects the factoids from interference when the knowledge is stored earlier (lower mean) and diversified (larger STD) in the layers.

**REMIX enables better knowledge manipulation.** Recent work has shown that manipulating learned knowledge is challenging especially during fine-tuning (Allen-Zhu & Li, 2024; 2025). We design two templates to evaluate: 1) Selective Recall and 2) Recall & Manipulate. For selective recall, the model that has memorized the factoids "`X: A, Y: B`" needs to answer the question "`Here are two keys:  X and Y. What is the value of the first key?`" with `A`. For recall & manipulate, the model that has memorized the factoid "`XYZ: ABC`" needs to answer the question "`If the first character in the value of key XYZ is changed to Q, what is the new value of key?`" with "`QBC`". We show in Table 5 that even though knowledge manipulation remains extremely hard for fine-tuning, REMIX still enables better manipulation of learned knowledge than no mixing, especially on the selective recall template.

| | **Selective Recall** | | **Recall & Manipulate** | |
|---|---|---|---|---|
| | Factoid | Non-Factoid | Factoid | Non-Factoid |
| No Mixing | 0.7 | 8.6 | 0.2 | 1.3 |
| REMIX (R/-) | 1.9 | **29.1** | 0.8 | **2.3** |
| REMIX (R/K-Pile) | **11.2** | 8.8 | **3.4** | 1.8 |

Table 5: Knowledge manipulation accuracy on KVR. R = Random Word Sequence. KP = Knowledge Pile. REMIX improves knowledge manipulation over No Mixing.

**Can REMIX go beyond two stages?** We test REMIX after more training stages to assess its effectiveness beyond the main two-stage setting. Figure 5 shows the accuracy of the Key-Value Recall task when trained on the combination of WebQA, EntityQA, MATH, and UltraChat. We observe a severe degradation when the two consecutive stages are both memorization-intensive. When the two following data are both factoid tasks, the No Mixing baseline is able to retain 37.0% accuracy. In contrast, REMIX can largely enhance the model's ability to retain knowledge, and is robust after two stages of training, leading at least 30% accuracy above the baseline across the board.

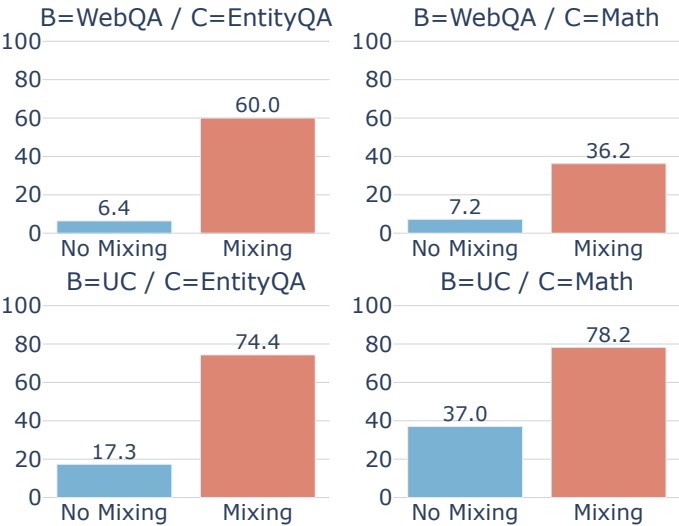

Figure 5: 3-stage continual memorization setting. $B = *$ refers to the stage 2 task, and $C = *$ refers to the stage 3 task. y-axis refers the accuracy (%) on Key-Value Recall. We use Random mixing at stage 1, K-Pile mixing at stage 2 for WebQA, No Mixing at stage 2 for UltraChat (UC), K-Pile mixing at stage 3 for EntityQA, and No Mixing for MATH at stage 3.

**Other ablation and analysis.** We provide extensive ablations and analysis on the effects of different mixing data, mixing data lengths, ablation of mixing ratio, and REMIX's impact on downstream tasks in §B.7.

## 6 Related Work

**Continual learning.** Continual learning has been the subject of investigation since early research on connectionist models, which identified *catastrophic forgetting* as a fundamental challenge (McCloskey & Cohen, 1989; Ratcliff, 1990). Many methods have been proposed for mitigating forgetting in continual learning. The simplest approach involves maintaining a memory of examples from previous tasks and replaying them during subsequent training (e.g., Robins, 1995; Chaudhry et al., 2019; Shin et al., 2017). Other methods involve regularization techniques that preserve important weights (e.g., Kirkpatrick et al., 2017; Ke et al., 2023) or reduce the divergence between model predictions (Li & Hoiem, 2017). One group of methods project the gradient for a new task to be orthogonal to the gradients from previous tasks, with the aim of reducing interference between tasks (Lopez-Paz & Ranzato, 2017; Farajtabar et al., 2020). A number of studies have attempted to characterize the relationship between task similarity and forgetting, empirically and theoretically (Ramasesh et al., 2021; Lee et al., 2021; Evron et al., 2022). In this paper, we restrict the class of approaches to those that do not change model weights, e.g., via regularization.

**Memorization and forgetting in LMs.** In the context of LMs, many prior works have investigated the factors that influence memorization during pre-training (Tirumala et al., 2022; Carlini et al., 2023; Mallen et al., 2023; Jagielski et al., 2023). In particular, prior work has observed that instruction tuning can lead to

some degradation on general NLP tasks, which has been called an "alignment tax" (Ouyang et al., 2022; Bai et al., 2022). Ouyang et al. (2022) find that this alignment tax can be partly mitigated by mixing pre-training data into the alignment data, and Luo et al. (2023a) find that LMs forget less when the instruction-tuning data is more diverse. Kotha et al. (2024) find that fine-tuning LMs leads to bigger performance degradation on tasks that are more similar to the fine-tuning task (as measured by likelihood under the learned fine-tuning distribution). See Shi et al. (2024) and Wu et al. (2024) for more extensive surveys of continual learning in the context of LMs.

**Fine-tuning on unfamiliar facts.** Our work builds on several recent observations about the effect of fine-tuning an LM on unfamiliar facts. Kang et al. (2025) find that fine-tuning LMs on unfamiliar examples (questions that the LM cannot answer correctly via few-shot prompting) leads the model to "hallucinate" plausible-sounding but incorrect answers to unfamiliar test examples. Similarly, Gekhman et al. (2024) find that unknown examples take longer to learn, and learning unknown examples leads to more hallucination. These studies highlight the difficulty of encoding new facts into a model during fine-tuning. Instead of directly learning the facts, Jang et al. (2022) and Seo et al. (2024) study the setting where the facts are embedded in the corpora and need to be learned continually. Yang et al. (2025) propose to address this challenge by generating synthetic data for continual pretraining. This approach can be motivated by mechanistic studies (Allen-Zhu & Li, 2024; 2025), which have found that knowledge extraction is possible only when information appears in diverse forms in the training data (e.g. paraphrases), which leads models to encode information more effectively for later extraction.

**Model editing and unlearning.** Our work is also related to a line of research aimed at explicitly modifying facts that are encoded in an LLM—for example, to update information about entities to reflect changes in the world (e.g., Zhu et al., 2020; Mitchell et al., 2022; Meng et al., 2022; 2023). Studies have shown that these methods can update individual facts, but do not lead to consistent changes about all of the implications of these updates (Zhong et al., 2023; Cohen et al., 2024). A related line of work has investigated whether specific information can be deliberately removed from neural networks (e.g., Graves et al., 2021; Zhang et al., 2023). Our focus in this paper is on introducing new information while retaining existing knowledge, rather than modifying or erasing existing knowledge.

# 7 Conclusion

In this paper, we formalize finetuning a language model with factual knowledge in the *continual memorization* framework. In contrast to continual learning, which focuses on general capabilities, we focus on the specific challenges inherent to finetuning to memorize knowledge. Through careful experiments, we establish that finetuning on factoid data causes the most severe forgetting on the memorized factoids from previous stages of finetuning. We then evaluate experience replay methods that are often used in continual learning and find that they do not satisfactorily revive forgotten factoids. To address the issue of forgetting, we propose a surprisingly effective strategy REMIX. By mixing random word sequences or generic pretraining data into different stages of training, REMIX outperforms replay-based methods and other baselines in our experiments despite not using any factoids from the original set in its mixing process. Finally, we analyze REMIX using Logit Lens and ablation studies to find that it teaches the model to change where it stores facts—moving it to earlier layers or diversifying the knowledge storage location. Studying the continual memorization problem opens up many new directions for future research. For example, future work may explore REMIX and similar approaches to ensure that safety-tuning is not easily undone by further finetuning. Its efficacy poses interesting questions about the dynamics of memorization in language models, which we are excited to see investigated in future work.

## Acknowledgments

We thank Alexander Wettig, Howard Yen, Tianyu Gao, Mengzhou Xia, and Yihe Dong for their valuable feedback on the manuscript. This research is supported by the National Science Foundation (IIS-2211779), a Sloan Research Fellowship, and Cisco Research.

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

# A   Derivations for Forgetting, Replay, and REMIX

## A.1   Forgetting in Continual Memorization

We give a formulation of when forgetting happens and how random and generic data mixing (REMIX) can mitigate forgetting.

We aim to analyze how mixing data during training affects memorization. Assume access to the *mixing dataset* $D_M$ while learning either $D_A$ or $D_B$ – training on $D'_A = D_A \cup D_M$ at stage 1 and converges to $\theta'_A$ or $D'_B = D_B \cup D_M$ at stage 2 and converges to $\theta'_B$. Our goal is to examine under what condition does the following occur:

$$\mathcal{L}(\theta_B; \mathcal{D}_A) > \mathcal{L}(\theta'_B; \mathcal{D}_A),$$

which means that through mixing, the final model $\theta'_B$ achieves a lower loss under $D_A$ than $\theta_B$.

We can track the progression of the model with the following stages:

$$\theta_A = \theta_0 - \eta\nabla\mathcal{L}(\theta_0; \mathcal{D}_A) \quad \text{(Stage 1; no mixing)}$$
$$\theta_B = \theta_A - \eta\nabla\mathcal{L}(\theta_A; \mathcal{D}_B) \quad \text{(Stage 2; no mixing)}$$

Note that this is a simplification of the actual optimization process as the *local one-step gradient* may point in a different direction from the final parameter difference $(\theta_A - \theta_0)$. We use $\nabla\mathcal{L}(\theta; D)$ to represent the conceptual overall direction for model $\theta$ to point to the low loss region of data $D$. The goal can be expressed as the difference:

$$
\begin{aligned}
\Delta &= \mathcal{L}(\theta_B; \mathcal{D}_A) - \mathcal{L}(\theta'_B; \mathcal{D}_A) \\
&= \Big( \mathcal{L}(\theta_A; \mathcal{D}_A) + (\theta_B - \theta_A)^T \nabla\mathcal{L}(\theta_A; \mathcal{D}_A) + \underbrace{R_1}_{\text{Higher-Order Terms}} \Big) \\
&\quad - \Big( \mathcal{L}(\theta'_A; \mathcal{D}_A) + (\theta'_B - \theta'_A)^T \nabla\mathcal{L}(\theta'_A; \mathcal{D}_A) + \underbrace{R_2}_{\text{Higher-Order Terms}} \Big) \\
&= \Big( \mathcal{L}(\theta_A; \mathcal{D}_A) - \eta\nabla\mathcal{L}(\theta_A; \mathcal{D}_B)^T \nabla\mathcal{L}(\theta_A; \mathcal{D}_A) \Big) \\
&\quad - \Big( \mathcal{L}(\theta'_A; \mathcal{D}_A) - \eta\nabla\mathcal{L}(\theta'_A; \mathcal{D}_B \cup D_M)^T \nabla\mathcal{L}(\theta'_A; \mathcal{D}_A) \Big) + (R_1 - R_2) \\
&= \underbrace{\mathcal{L}(\theta_A; \mathcal{D}_A) - \mathcal{L}(\theta'_A; \mathcal{D}_A)}_{\Delta_1} \\
&\quad + \underbrace{\eta\Big( \nabla\mathcal{L}(\theta'_A; \mathcal{D}_B \cup D_M)^T \nabla\mathcal{L}(\theta'_A; \mathcal{D}_A) - \nabla\mathcal{L}(\theta_A; \mathcal{D}_B)^T \nabla\mathcal{L}(\theta_A; \mathcal{D}_A) \Big)}_{\Delta_2} \\
&\quad + \underbrace{(R_1 - R_2)}_{\Delta_3}
\end{aligned}
$$

We assume that the first two terms $\Delta_1, \Delta_2$ are the main source contributing to forgetting and ignore the higher-order terms.

## A.2   Replay

In the replay scenario, the *mixing data $D_M$* is a subset of $D_A$. We denote the $r\%$ subset of $D_A$ as $D^r_A$. With $D_M = D^r_A$, we can assert that $\Delta_1 \approx 0$ since the converged model should obtain the same loss under $D_A$ and

$D_A \cup D_A^r$. The second term $\Delta_2 = \nabla\mathcal{L}(\theta_A'; D_B \cup D_A^r)^T \nabla\mathcal{L}(\theta_A'; D_A) - \nabla\mathcal{L}(\theta_A; D_B)^T \nabla\mathcal{L}(\theta_A; D_A) > 0$.

$$\begin{aligned}
\Delta_2 &= \nabla\mathcal{L}(\theta_A'; D_B \cup D_A^r)^T \nabla\mathcal{L}(\theta_A'; D_A) - \nabla\mathcal{L}(\theta_A; D_B)^T \nabla\mathcal{L}(\theta_A; D_A) \\
&\approx \left( \nabla\mathcal{L}(\theta_A'; D_B \cup D_A^r) - \nabla\mathcal{L}(\theta_A; D_B) \right)^T \nabla\mathcal{L}(\theta_A; D_A) \\
&> 0
\end{aligned}$$

### A.3 REMIX

**Mixing at stage 1: $D_A' = D_A \cup D_M$.** $\Delta_1 \approx 0$ due to convergence in either no mixing or mixing training scenarios. We turn to analyzing $\Delta_2$. The term $\nabla\mathcal{L}(\theta_A'; D_A) \approx \nabla\mathcal{L}(\theta_A; D_A) + H_A(\theta_A' - \theta_A)$ and $\nabla\mathcal{L}(\theta_A'; D_B) \approx \nabla\mathcal{L}(\theta_A; D_B) + H_B(\theta_A' - \theta_A)$, where $H_A$ is the Hessian of $\nabla\mathcal{L}(\theta; D_A)$ at $\theta = \theta_A$, and $H_B$ is the Hessian of $\nabla\mathcal{L}(\theta; D_B)$ at $\theta = \theta_B$. With mixing at stage 1, we have $\theta_A' = \theta_0 - \eta\nabla\mathcal{L}(\theta_0; D_A \cup D_M)$, which gives us $\theta_A' - \theta_A = \eta(\nabla\mathcal{L}(\theta_0; D_A) - \nabla\mathcal{L}(\theta_0; D_A \cup D_M)) = -\eta\nabla\mathcal{L}(\theta_0; D_M)$.

$$\begin{aligned}
\Delta_2 &= \eta\Big( \nabla\mathcal{L}(\theta_A'; D_B)^T \nabla\mathcal{L}(\theta_A'; D_A) - \nabla\mathcal{L}(\theta_A; D_B)^T \nabla\mathcal{L}(\theta_A; D_A) \Big) \\
&= \eta\Big( \big( \nabla\mathcal{L}(\theta_A; D_B) + H_B(\theta_A' - \theta_A) \big)^T \big( \nabla\mathcal{L}(\theta_A; D_A) + H_A(\theta_A' - \theta_A) \big) \\
&\quad - \nabla\mathcal{L}(\theta_A; D_B)^T \nabla\mathcal{L}(\theta_A; D_A) \Big) \\
&= \eta\Big( \big( \nabla\mathcal{L}(\theta_A; D_B) + H_B(-\eta\nabla\mathcal{L}(\theta_0; D_M)) \big)^T \big( \nabla\mathcal{L}(\theta_A; D_A) + H_A(-\eta\nabla\mathcal{L}(\theta_0; D_M)) \big) \\
&\quad - \nabla\mathcal{L}(\theta_A; D_B)^T \nabla\mathcal{L}(\theta_A; D_A) \Big) \\
&= -\eta^2 \nabla\mathcal{L}(\theta_A; D_B)^T H_A \nabla\mathcal{L}(\theta_0; D_M) - \eta^2 \nabla\mathcal{L}(\theta_A; D_A)^T H_B \nabla\mathcal{L}(\theta_0; D_M) \\
&\quad + \eta^3 \nabla\mathcal{L}(\theta_0; D_M)^T H_B H_A \nabla\mathcal{L}(\theta_0; D_M)
\end{aligned}$$

We analyze the three terms under the assumption that $H_A$, $H_B$, and $H_B H_A$ are positive semi-definite. If the distributions for $D_M$ and $D_B$ are *uncorrelated*, then in expectation $\mathbb{E}[\nabla\mathcal{L}(\theta_A; D_B)^T H_A \nabla\mathcal{L}(\theta_0; D_M)] = 0$. The same holds for $D_M$ and $D_A$. And the last term will be positive, contributing to $\Delta_2$ and thus mitigate forgetting. Note that the norm $||\nabla\mathcal{L}(\theta_0; D_M)||$ and the eigenvalues of the Hessians $H_A$ and $H_B$ are not bounded, which may be large and compensate for the leading $\eta^3$. If we assume that mixing $D_M$ does not drift the parameters away too far, making $||\nabla\mathcal{L}(\theta_A'; D_B) - \nabla\mathcal{L}(\theta_A; D_B)||_2^2 < L_1$, and $||\nabla\mathcal{L}(\theta_A'; D_A) - \nabla\mathcal{L}(\theta_A; D_A)||_2^2 < L_2$, where $L_1, L_2 \in \mathbb{R}$, we can expect the contribution to the $\Delta_2$ term comes from the change in the angle.

**Mixing at stage 2: $D_B' = D_B \cup D_M$.** With no mixing in stage 1, we have $A' = A$. Therefore, the first term $\Delta_1 = \mathcal{L}(\theta_A; D_A) - \mathcal{L}(\theta_A'; D_A) = 0$ since $D_A' = D_A$. We can also express:

$$\begin{aligned}
\Delta_2 &= \eta\Big( \nabla\mathcal{L}(\theta_A; D_B \cup D_M)^T \nabla\mathcal{L}(\theta_A; D_A) - \nabla\mathcal{L}(\theta_A; D_B)^T \nabla\mathcal{L}(\theta_A; D_A) \Big) \\
&= \eta\Big( \beta_1 \nabla\mathcal{L}(\theta_A; D_B)^T \nabla\mathcal{L}(\theta_A; D_A) + \beta_2 \nabla\mathcal{L}(\theta_A; D_M)^T \nabla\mathcal{L}(\theta_A; D_A) \\
&\quad - \nabla\mathcal{L}(\theta_A; D_B)^T \nabla\mathcal{L}(\theta_A; D_A) \Big) \\
&= \eta\Big( \beta_1 \nabla\mathcal{L}(\theta_A; D_M) - (1 - \beta_2)\nabla\mathcal{L}(\theta_A; D_B) \Big)^T \nabla\mathcal{L}(\theta_A; D_A),
\end{aligned}$$

where $\beta_1, \beta_2 \in [0, 1]$.

Consequently, the condition for forgetting mitigation requires $\nabla\mathcal{L}(\theta_A; D_M)^T \nabla\mathcal{L}(\theta_A; D_A) > \frac{1 - \beta_2}{\beta_1} \nabla\mathcal{L}(\theta_A; D_B)^T \nabla\mathcal{L}(\theta_A; D_A)$. This condition posits that mixing data can reduce forgetting as long as it aligns with the original data $D_A$ more than $D_B$. When $D_A$ and $D_B$ are already pointing in drastically

opposite directions, making the term $\nabla\mathcal{L}(\theta_A; D_B)^T\nabla\mathcal{L}(\theta_A; D_A)$ negative, the mixing has a higher chance to lower $\Delta_2$. On the other hand, if $\nabla\mathcal{L}(\theta_A; D_B)^T\nabla\mathcal{L}(\theta_A; D_A)$ is positive, it is harder for mixing to mitigate forgetting.

# B  Supplementary Results

## B.1  Main Results with Standard Deviation

We provide our main results with error bars over 3 runs in Figure 6 where stage 2 training uses factoid datasets and Figure 7 where stage 2 training uses non-factoid datasets.

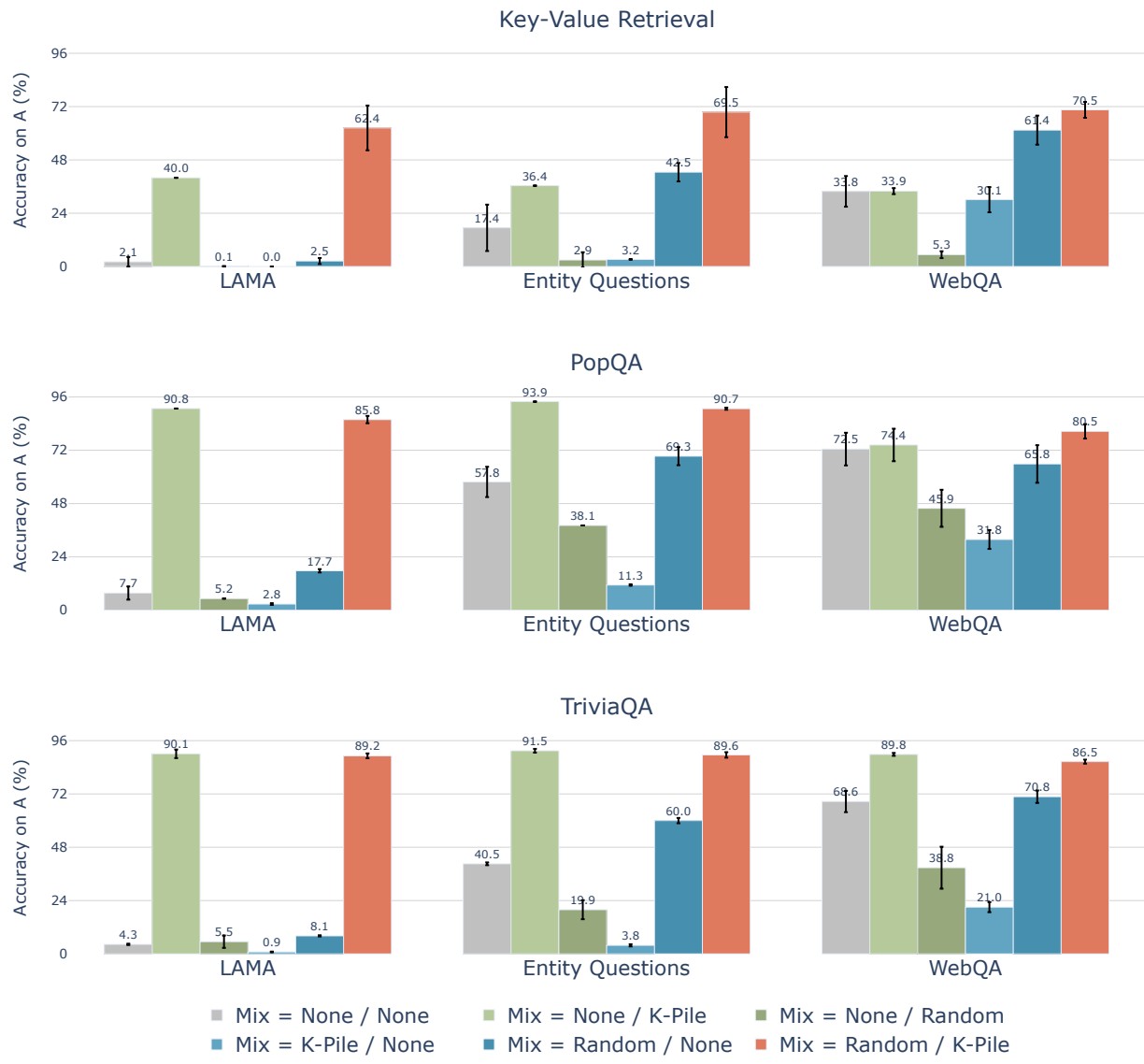

Figure 6: Accuracies of different combinations of $D_A$ (rows) against $D_B$ (columns) over 3 seeds on the factoid datasets. Legends show different mixing combinations $M_A/M_B$ where $M_A$ is the mixing data used in stage 1 and $M_B$ is the mixing data used in stage 2. The performance (y-axis) is measured on $D_A$.

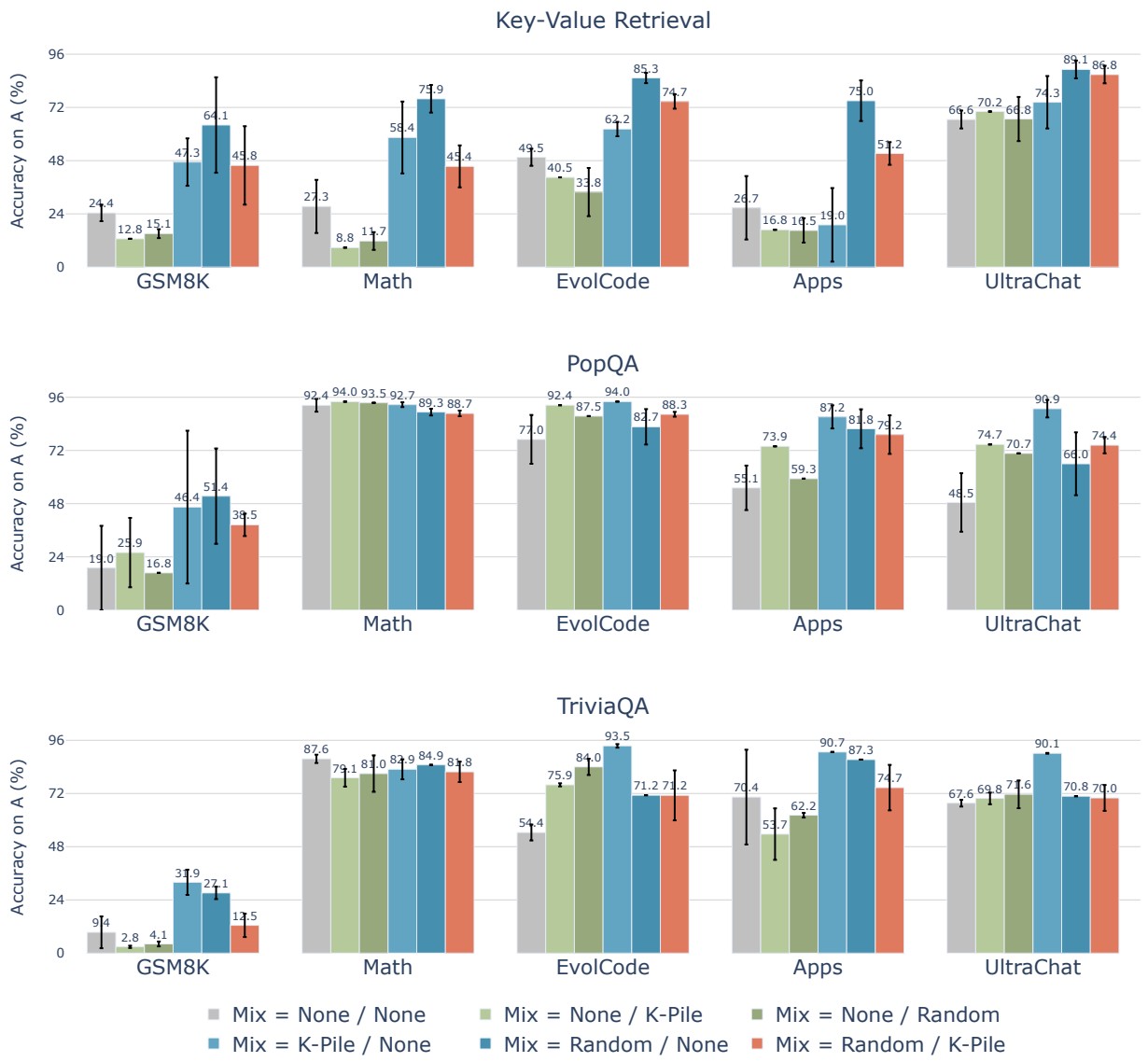

Figure 7: Accuracies of different combinations of $D_A$ (rows) against $D_B$ (columns) over 3 seeds on the non-factoid datasets. Legends show different mixing combinations $M_A/M_B$ where $M_A$ is the mixing data used in stage 1 and $M_B$ is the mixing data used in stage 2. The performance (y-axis) is measured on $D_A$.

## B.2 Stage 2 Performance

For factoid datasets, the goal is full memorization of the trained examples. For non-factoid datasets, we also train to near perfect training accuracy (the overfitting regime) since we aim to assess the maximum disruption that training can cause. We show the corresponding accuracy in Table 6 of the main paper (for non-factoid data, we only show the ones where accuracy can be calculated). We show the representative strategies: No Mixing and Random / -. Results show near perfect accuracy for $D_B$, indicating that learning does not hinder performance. The only exception is KVR (No Mixing)— this further highlights the benefit of REMIX in facilitating learning. With REMIX, all training reaches over 95% accuracy.

We also provide test accuracies in Table 7 for the non-factoid datasets where separate test sets are available. Note that for factoid datasets, each example is an isolated fact to be memorized exactly, therefore the notion of generalization does not apply. We observe that for KVR, REMIX improves generalization noticeably

|  | LAMA | EntQA | WebQA | GSM8K | MATH | APPS |
|---|---|---|---|---|---|---|
| KVR (No Mixing) | 95.6 | 99.8 | 99.3 | 87.3 | 74.1 | 5.4 |
| KVR (Rand / -) | 96.0 | 98.3 | 99.3 | 99.4 | 99.3 | 98.1 |
| PopQA (No Mixing) | 95.1 | 98.9 | 98.9 | 98.8 | 97.7 | 95.7 |
| PopQA (Rand / -) | 95.6 | 99.0 | 98.9 | 98.9 | 98.1 | 95.8 |
| TriviaQA (No Mixing) | 95.8 | 98.7 | 98.4 | 98.8 | 98.3 | 95.0 |
| TriviaQA (Rand / -) | 95.5 | 98.8 | 98.9 | 98.6 | 98.2 | 95.8 |

Table 6: Training set accuracy of $D_B$ datasets after stage 2 training. All datasets are trained to full convergence to induce maximal forgetting in $D_A$.

|  | GSM8K (train/test) | MATH (train/test) | APPS (train/test) |
|---|---|---|---|
| KVR (No Mixing) | 87.3 / 19.1 | 74.1 / 5.1 | 5.4 / 0.5 |
| KVR (Rand / -) | 99.4 / **27.1** | 99.3 / **8.4** | 98.1 / **4.5** |
| PopQA (No Mixing) | 98.8 / **27.6** | 97.7 / **8.5** | 95.7 / **2.7** |
| PopQA (Rand / -) | 98.9 / 26.5 | 98.1 / 7.1 | 95.8 / 0.7 |
| TQA (No Mixing) | 98.8 / 27.2 | 98.3 / 8.6 | 95.0 / 1.2 |
| TQA (Rand / -) | 98.6 / **27.4** | 98.2 / **8.8** | 95.8 / **2.7** |

Table 7: Training and test set accuracy of $D_B$ datasets after stage 2 training. All datasets are trained to full convergence to induce maximal forgetting in $D_A$. For KVR and TQA, REMIX improves generalization over the no mixing baseline.

across all tasks. For PopQA and TriviaQA, REMIX's generalization ability is close to No Mixing (within 2 point range).

We use only 2000 examples for all datasets during training and deliberately overfit on to induce maximal forgetting on, so the test performance level is expected. We would like to emphasize that overfitting on non-factoid is necessary for the purpose of our goal to induce the forgetting pattern in Table 2, which allows us to stress test retention of $D_A$, or otherwise the forgetting is much less pronounced for non-factoid to begin with.

### B.3 Mistral Results

We report the complete results of Mistral-7B-v0.3 in Table 8.

### B.4 Replay Results

We report the full replay results in Table 9. Even though replay reduces more forgetting across the board, especially when we increase the ratio $r$, the replay-based method does not effectively mitigate forgetting in the factoid knowledge dataset.

### B.5 Forgetting in Familiar Factoid Instances

We also investigate whether REMIX can retain the memorization of familiar factoid instances after directly fine-tuning on both factoid and non-factoid data in stage 2. After fine-tuning in stage 2, we evaluated the familiar instances from the factoid dataset $D_A$. The evaluation results for Llama-3-8B are shown in Table 10. We observe that mixing Knowledge-Pile, Arxiv-Pile, and FineWeb with factoid data in stage 2 helps mitigate the forgetting of familiar factoid instances for both Llama-3-8B and Mistral-7B-v0.3, aligning with the results in Figure 6.

### B.6 Probing Results

| | Factoid | | | | | Non-Factoid | | | | |
|---|---|---|---|---|---|---|---|---|---|---|
| | LAMA | EQ | WQ | **Avg** | GSM8K | MATH | EC | APPS | UC | **Avg** |
| **Key-Value Recall** | | | | | | | | | | |
| No Mixing | 0.1 | 15.4 | 29.6 | 15.0 | 4.8 | 1.5 | 12.7 | 13.1 | 51.9 | 16.8 |
| Random / K-Pile | 47.5 | 44.1 | 39.0 | **43.5** | 60.1 | 39.1 | 52.9 | 54.8 | 81.0 | **57.0** |
| **PopQA** | | | | | | | | | | |
| No Mixing | 66.9 | 92.3 | 89.6 | 82.9 | 96.9 | 96.8 | 96.9 | 96.9 | 96.7 | **96.8** |
| Random / K-Pile | 90.5 | 92.3 | 89.0 | **90.6** | 91.7 | 91.6 | 91.8 | 91.9 | 91.3 | 91.7 |
| **TriviaQA** | | | | | | | | | | |
| No Mixing | 71.6 | 86.4 | 91.5 | **83.2** | 4.8 | 99.0 | 95.9 | 79.9 | 97.0 | **75.3** |
| Random / K-Pile | 77.0 | 81.5 | 83.1 | 80.5 | 1.6 | 91.1 | 95.3 | 97.7 | 90.7 | **75.3** |

Table 8: REMIX results for Mistral-7B-v0.3. We compare the No Mixing baseline to REMIX that mixes with Random Word Sequence at stage 1 and mixes with Knowledge Pile at stage 2. (EQ = EntityQA, WQ = WebQA, EC = EvolCode, and UC = UltraChat.)

| | LAMA | EntityQA | WebQA | GSM8K | Math | EvolCode | Apps | UltraChat |
|---|---|---|---|---|---|---|---|---|
| **Key-Value Recall** | | | | | | | | |
| Replay ($r = 0.00$) | 2.2 | 17.5 | 34.1 | 26.4 | 27.5 | 50.0 | 30.0 | 66.7 |
| Replay ($r = 0.01$) | 13.7 | 37.1 | 54.2 | 71.0 | 69.7 | 73.2 | 73.8 | 81.9 |
| Replay ($r = 0.05$) | 6.3 | 45.8 | 72.6 | 77.0 | 75.9 | 76.7 | 80.1 | 88.9 |
| Replay ($r = 0.1$) | 13.2 | 33.3 | 78.2 | 80.3 | 85.0 | 76.5 | 86.7 | 91.1 |
| **PopQA** | | | | | | | | |
| Replay ($r = 0.00$) | 15.7 | 64.3 | 78.6 | 33.6 | 93.5 | 80.5 | 63.2 | 53.7 |
| Replay ($r = 0.01$) | 12.0 | 66.0 | 75.3 | 94.4 | 95.1 | 95.7 | 90.8 | 87.6 |
| Replay ($r = 0.05$) | 27.4 | 64.4 | 84.5 | 95.9 | 95.2 | 95.4 | 95.9 | 95.3 |
| Replay ($r = 0.1$) | 46.6 | 64.0 | 83.8 | 96.1 | 96.0 | 95.7 | 96.3 | 95.7 |
| **TriviaQA** | | | | | | | | |
| Replay ($r = 0.00$) | 7.8 | 48.4 | 76.8 | 57.6 | 91.0 | 59.5 | 75.6 | 73.5 |
| Replay ($r = 0.01$) | 7.5 | 51.8 | 72.0 | 66.8 | 90.6 | 93.3 | 74.2 | 84.0 |
| Replay ($r = 0.05$) | 25.7 | 57.0 | 77.8 | 88.9 | 94.0 | 93.7 | 94.4 | 92.0 |
| Replay ($r = 0.1$) | 34.9 | 57.9 | 80.7 | 93.0 | 95.5 | 95.4 | 95.2 | 93.0 |

Table 9: Replay accuracy on $D_A$ (rows) after training on the unfamiliar factoid and non-factoid datasets $D_B$ (columns) at four replay ratios $[0.0, 0.01, 0.05, 0.1]$. Lowest number among the compared rations are underlined. The results are based on Llama-3-8B.

### B.7 Ablations

**Ablating mixing data length.** Figure 10 shows the effect of sequence length when using Random Word Sequences and Knowledge Pile for mixing. We observe that longer Random Word Sequences hurt the performance, highlighting the risk of incorporating wildly out of distribution data. On the other hand, Knowledge Pile also saturates after 50 words, indicating the limits of the generic data. The ablation also affirms that the role of the mixing data serves as a way to manipulate the memorization dynamics as opposed to providing extra information.

**Effect of mixing ratio.** We show in Figure 11 the model's KVR performance under varying mixing ratio across all stage 2 tasks. We observe that stage 2 mixing is particularly sensitive to the increase of mixing ratio. On the other hand, stage 1 mixing enjoys less decrease or even increase in performance as the mixing ratio goes up, suggesting a different memorization dynamics than stage 1.

|  | LAMA | EntityQA | WebQA | GSM8K | Math | EvolCode | Apps | UltraChat |
|---|---|---|---|---|---|---|---|---|
| **PopQA** | | | | | | | | |
| No Mixing | 27.3 | 24.4 | 39.1 | 13.0 | 18.3 | 36.3 | 7.4 | 46.9 |
| K-Pile | 56.0 | 52.1 | 46.6 | 4.1 | 4.8 | 19.5 | 10.3 | 15.8 |
| A-Pile | 65.1 | 60.4 | 52.7 | 9.2 | 3.2 | 26.5 | 21.8 | 19.3 |
| Random | 24.9 | 27.9 | 29.1 | 7.5 | 6.0 | 25.4 | 2.8 | 18.4 |
| FineWeb | 54.9 | 54.4 | 51.3 | 6.6 | 5.2 | 29.1 | 30.0 | 18.4 |
| **TriviaQA** | | | | | | | | |
| No Mixing | 16.5 | 20.7 | 40.4 | 24.7 | 26.7 | 52.9 | 21.9 | 56.6 |
| K-Pile | 55.9 | 57.5 | 50.3 | 11.0 | 6.8 | 28.4 | 20.9 | 23.4 |
| A-Pile | 66.4 | 65.9 | 56.6 | 13.0 | 2.8 | 34.8 | 33.5 | 25.8 |
| Random | 14.4 | 26.5 | 26.5 | 14.0 | 7.5 | 21.8 | 13.4 | 27.5 |
| FineWeb | 56.6 | 57.6 | 52.6 | 13.0 | 6.0 | 38.9 | 56.9 | 17.7 |

Table 10: Accuracy on the *familiar* factoid datasets (rows) after training on the factoid and non-factoid datasets (columns) with different mixing data (mixed at stage 2). The results are based on Llama-3-8B.

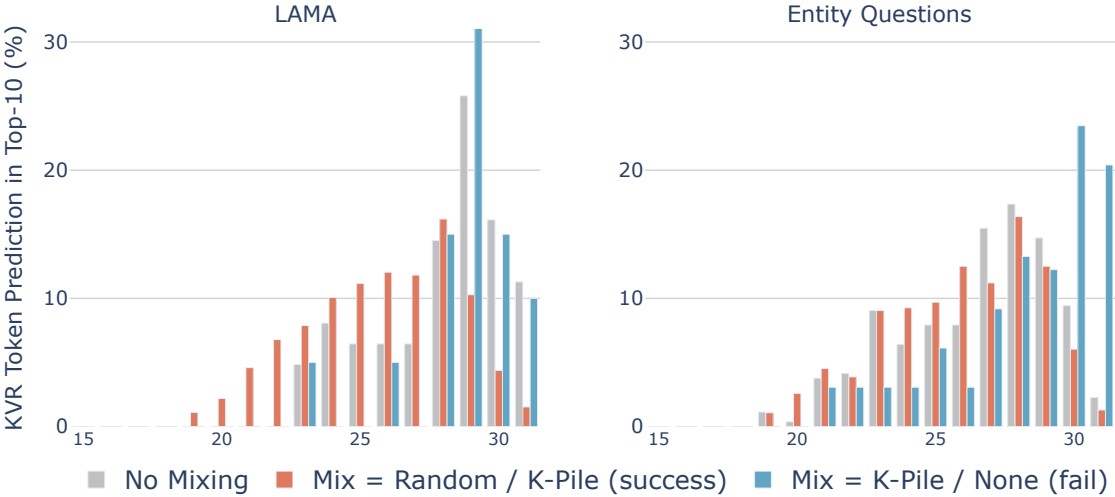

Figure 8: Probing of the Key-Value Recall task. x-axis: layer index. y-axis: the normalized frequency of the correct token occurring in the top-10 tokens probed at each layer.

**Impact on downstream performance.** Intuitively, adding random word sequences might risk disrupting capabilities in other domains. We evaluate the model's performance on MMLU Hendrycks et al. (2021a) shown in Table 11. We observe REMIX maintains better performance across the board compared to no mixing.

|  | KVR | PopQA | TriviaQA |
|---|---|---|---|
| No Mixing | 18.5 | 19.0 | 17.5 |
| REMIX (R / K-Pile) | **24.1** | **27.0** | **21.5** |

Table 11: Accuracy on MMLU. We compare the No Mixing baseline to REMIX, which mixes with Random Word Sequence (R) at stage 1 and with Knowledge Pile (K-Pile) at stage 2.

**Effect of different mixing data.** We investigate how the choice of the mixing data impacts the results for factoid-tasks. Figure 12 shows no difference between Knowledge Pile and other generic mixing data such

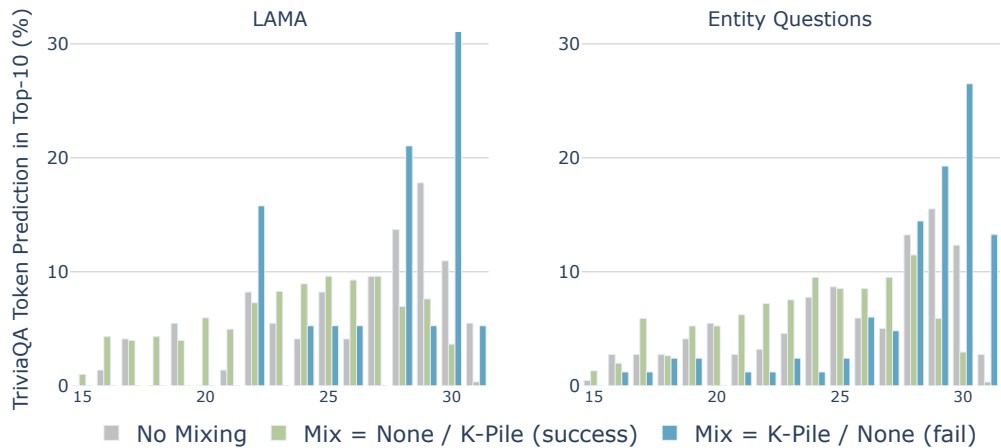

Figure 9: Probing of the TriviaQA task. x-axis: layer index. y-axis: the normalized frequency of the correct token occurring in the top-10 tokens probed at each layer.

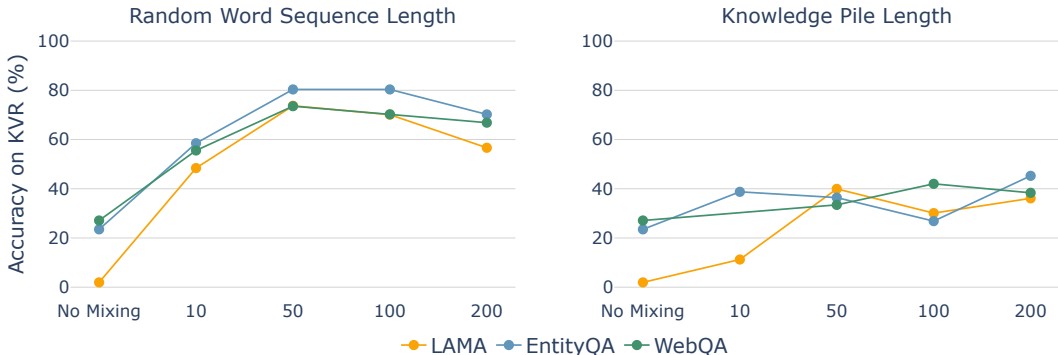

Figure 10: y-axis is the accuracy (%) on Key-Value Recall of varying sequence length with the mixing datasets. Top: Random Word Sequence (mixed at stage 1). Bottom: Knowledge Pile (mixed at stage 2).

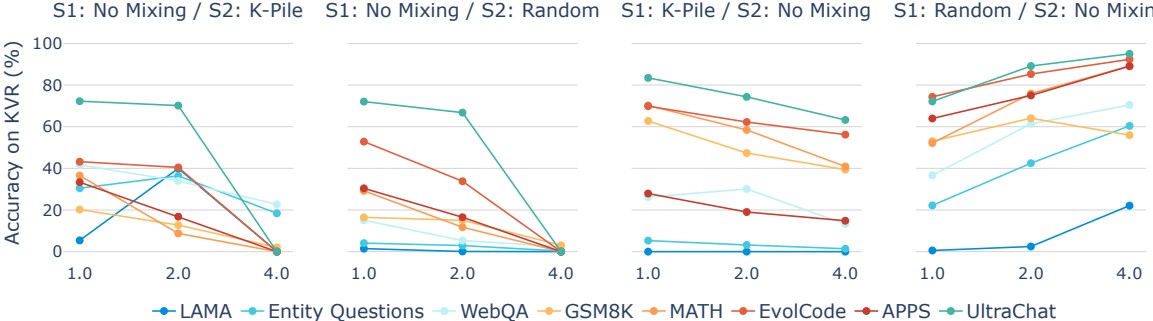

Figure 11: Mixing ratio ablation. x-axis indicates the ratio of the mixing data against the training data. y-axis indicates the accuracy (%) on Key-Value Recall. The two left-most plots are both stage 2 mixing (S2) and the right-most two are both stage 1 mixing (S1).

as ArXiv Pile and FineWeb. This affirms that the effectiveness of REMIX does not rely on Knowledge Pile's potential distributional overlap with memorization-intensive tasks.

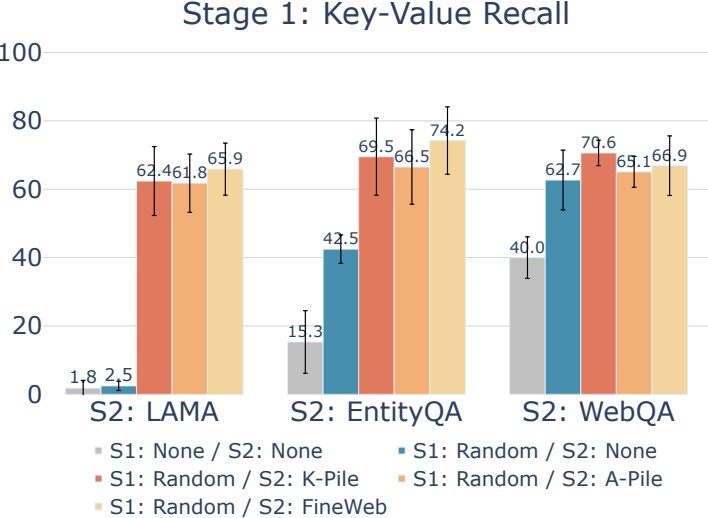

Figure 12: Comparison between Knowledge Pile and other generic mixing data sources: ArXiv Pile and FineWeb on KVR. y-axis indicates the accuracy (%) on KVR.

### B.8 Training Details

For all experiments with Llama-3-8B, we average the results over three seeds and use a learning rate of 5e-5. For all experiments with Mistral-7B-v0.3, we use a learning rate of 1e-5. For experiments measuring forgetting of familiar factoid datasets, we use a batch size of 128. For the rest of the experiments, we set the batch size to 32. Additionally, different stopping conditions are applied for the different factoid datasets: for the KVR task, we use a fixed number of epochs (20), while for other factoid tasks, training stops when the loss drops below 0.0001. We provide our training prompt in §B.9.

### B.9 Dataset Examples and Prompts

We provide examples of the stage 1 factoid datasets $D_A$ and the mixing datasets $D_M$. Since stage 2 non-factoid datasets are standard instruction tuning datasets, we omit these examples in the following sections.

#### B.9.1 Factoid dataset ($D_A$) Examples

1. **Key-Value Recall**
   Input:    The value of key e6395973 is?
   Target:   8219acf2

2. **PopQA**
   Input:    Question:  What is New Lands's author?  The answer is:
   Target:   Charles Fort

3. **TriviaQA**
   Input:    Question:  Which city does David Soul come from?  The answer is:
   Target:   Chicago

#### B.9.2 Mixing data ($D_M$) Examples

1. **Knowledge-Pile**
   Input:  Complete the following partial passage:  Processing hyperspectral images
   allows you to decode images and recognize objects in the scene on the base of

```
analysis of spectrums.  In some problems, information about the spectra may not
be sufficient.  In this case, visualization of data sets may use, for object
recognition, by use additional non-formalized external attributes
```
Target:  (for example, indicating the relative position of objects).  Target
```
visualization is a visualization adapted to a specific task of application.
The method discussed in this chapter uses a way to visualize a measure of
similarity to the sample.  As a result of the transformation, the hyperspectral
(multichannel) image is converted [...]
```

2. **Arxiv-Pile**
   Input:  Complete the following partial passage:  -- abstract:  'The purpose of
   ```
   this article is to study the problem of finding sharp lower bounds for the norm of
   ```
   the product of polynomials in the ultraproducts of Banach spaces $(X_i)_\mathfrak{U}$.  We show
   ```
   that, under certain hypotheses, there is a strong relation between this problem
   and the same
   ```
   Target:  problem for the spaces $X_i$.'  address:  'IMAS-CONICET' author:  - Jorge
   ```
   Tomás Rodríguez title:  On the norm of products of polynomials on ultraproducts
   of Banach spaces -- Introduction ============ In this article we study the factor
   problem in the context of ultraproducts of Banach spaces.  This problem can be
   stated as [...]
   ```

3. **FineWeb**
   Input:  Complete the following partial passage:  *sigh* Fundamentalist community,
   ```
   let me pass on some advice to you I learned from the atheistic community:  If you
   have set yourself on fire, do not run.  Okay?  Okay??  Please?  Look, D, you had
   two months to say to Harvard in private emails, "I'm sorry, I shouldn't have been
   using
   ```
   Target:  that animation in my paid presentations.  I wont use it again.  I really
   ```
   do like 'Inner Life', though, and would love to use it in classroom presentations,
   from the BioVisions site, if that is acceptable." I sat here, for two months,
   waiting for that to happen, anything to happen, and [...]
   ```

4. **Random Word Sequence**
   Input:  Memorize the following random-string passage:  pliosaur bismuth
   ```
   assertoric decentralization emerse redemonstrate sleepwaker Coracias thirstland
   Stercorariinae Cytherean autobolide pergamentaceous ophthalmodynamometer
   tensify tarefitch educement wime cockneity holotype spreng justiciary unseparate
   ascogonial chirimen Styphelia emotivity heller hystazarin unthinkable Corinth
   vicianose incommunicative sorcerous lineograph dochmiacal heresiographer
   interrenal anes mercal embryogenic swoon diptote funniness unwreathed contection
   rhapsodical infolding colorature multifurcate
   ```
   Target:  pliosaur bismuth assertoric decentralization emerse redemonstrate
   ```
   sleepwaker Coracias thirstland Stercorariinae Cytherean autobolide pergamentaceous
   ophthalmodynamometer tensify tarefitch educement wime cockneity holotype spreng
   justiciary unseparate ascogonial chirimen Styphelia emotivity heller hystazarin
   unthinkable Corinth vicianose incommunicative sorcerous lineograph dochmiacal
   heresiographer interrenal anes mercal embryogenic swoon diptote funniness
   unwreathed contection rhapsodical infolding colorature multifurcate
   ```

## B.10  Dataset Details

We examine the strict overlap of knowledge entities between PopQA, TriviaQA, and the generic data used
for mixing.  By extracting knowledge entity pairs from the questions and target answers, we calculate the
exact overlap between these pairs.  The overlap percentage among PopQA, TriviaQA, and the generic data
is less than 1.3%.  Note that the entity overlap ratio calculated is an overestimate.  For example, parsing

the instance "Question: Behind Russia, what is the second largest country in Europe? Answer: Ukraine" results in entities like "Russia", "Europe", and "Ukraine". It's easy to find in generic corpora a text that mentions all the entities, yet the relation "the second largest" is not present. This is different from the replay data which is the exact instance that contains this relation. If we apply strict entity parsing rules and only extract tags such as PERSON, PRODUCT, ORG, then the overlap is close to zero ($\sim 0.1\%$).

### B.11   Caveats for Interpreting Table 1

We provide a more nuanced reading of Table 1. First, the factoid and non-factoid $D_B$'s differ along more dimensions beyond factoid vs. non-factoid, and we do not control for all the possible variables. Second, individual datasets do not always follow the aggregate trend. For example, GSM8K (non-factoid) induces severe forgetting on PopQA (19.0%) and TriviaQA (9.4%), comparable to or exceeding some factoid $D_B$'s. Conversely, WebQA (factoid) causes relatively mild forgetting on KVR (33.8%) and TriviaQA (68.6%). The overall pattern is broadly consistent with findings from the continual learning literature suggesting that catastrophic forgetting tends to be more pronounced when two tasks are similar, potentially causing greater interference (Farajtabar et al., 2020; Bennani et al., 2020; Doan et al., 2021). The trend warrants further controlled investigation to disentangle the effect of task type from other confounding factors.

