# OpenReview forum: "Continual Memorization of Factoids in Language Models"
_TMLR — Accepted by TMLR_

### Review · Reviewer_S9M8 · 2025-07-11

**Summary Of Contributions:**

This paper investigates the problem of continual memorization in language models (LMs), where models must retain factual knowledge across multiple fine-tuning stages. The authors show that standard fine-tuning leads to significant forgetting, especially for facts introduced later. The authors attribute this to suboptimal training dynamics and propose REMIX, a method that mixes in random or generic data during training to mitigate forgetting. REMIX outperforms replay-based and other continual learning baselines, and analysis reveals that it promotes more robust and distributed fact retention across model layers.

**Audience:**

Yes

**Claims And Evidence:**

Yes

**Requested Changes:**

1. There is a typo in Section 2.1 (line 4): the second $D$ should be $\mathcal{D}$. Also, consider using more distinguishable notations (e.g., $\hat{D}$) to differentiate $D$ and $\mathcal{D}$.
2. The factoid datasets $D_A$ are constructed by removing all familiar instances. This may be unrealistic. Real-world updates often include some overlap. It would be helpful to analyze whether including a small portion of familiar data affects the outcomes.
3. The interpretation of Table 1 is debatable. Since different rows correspond to different $(D_A, D_B)$ pairs, comparing absolute accuracy values may be misleading due to varying dataset difficulty. In fact, if only looking part of Table 1, we can even draw the opposite conclusion: the first two rows of the WQ column (factoid) is larger than the first two rows of APPS (non-factoid) column, which implies the forgetting is more severe in non-factoid datasets.
4. The mathematical formulation in Section 4.1 is unclear. Presenting the main claim as a formal proposition or theorem could improve clarity and rigor.
5. On Page 5 (third line from the bottom), the expression $L(\theta_B;D_A) - L(\theta_B;D_A)$ appears incorrect. It trivially evaluates to zero.
6. Since the REMIX dataset $D_M$ is comparable in size to $D_A$, the proposed approach substantially increases training volume. This likely leads to longer training time, but the paper does not discuss this trade-off.
7. The paper claims no overlap between factoid and mixing data (Page 6), yet Appendix B.10 reports up to 1.3% overlap. Given Figure 2 shows even 0.01 replay ratio improves accuracy, this overlap could be non-negligible and warrants more discussion.

**Strengths And Weaknesses:**

The paper presents a clear central idea and addresses an important topic: mitigating forgetting in language models. The experimental section is thorough. However, several parts of the methodology and analysis would benefit from further clarification.

---

> ### Author Response · Authors · 2025-11-21
> **Response to Reviewer S9M8**
>
> We thank the reviewer for the detailed comments and suggestions. We provide the response to the individual points below.
>
> **Using unfamiliar examples might be unrealistic**
> > The factoid datasets  are constructed by removing all familiar instances. This may be unrealistic. Real-world updates often include some overlap. It would be helpful to analyze whether including a small portion of familiar data affects the outcomes.
>
> The reason for separating the familiar and unfamiliar factoids is inspired by Kang et al., (2024) and Gekhman et al., (2024), where they found that training on unfamiliar instances can disrupt model behavior such as exacerbating hallucinations. We choose the unfamiliar factoids expecting that they will make forgetting patterns more apparent. In addition, the familiar factoids are learned during pretraining, which may have different forgetting characteristics than the unfamiliar factoids that are learned through fine-tuning in stage 1. We aim to disentangle these confounding factors in the main experiments.
> While we mainly investigate unfamiliar factoids, we also provide experiments on familiar factoids in Table 10 (Appendix B.7), which shows that mixing provides benefits in the familiar factoids setting as well.
>
>
> **Prompting the mathematical formulations in section 4.1 to proposition/theorem**
>
> Thank you for the suggestion. We have updated the section and formulated the descriptions into a proposition. Since the main evidence provided in the paper remains empirical, we defer the theoretical analysis in the appendix as a supplementary for establishing intuition for the proposed method.
>
>
> **Comparing absolute accuracy values may be misleading**
> > Interpretation of Table 1: comparing absolute accuracy values may be misleading due to varying dataset difficulty
>
> The accuracy reported represents the retention on the dataset $D_A$ after learning $D_B$. We first make sure $D_A$ is learned to 100% memorization in stage 1 before starting stage 2. We then train on $D_B$ to full convergence as shown in Table 6 (Appendix B.2). In this setting, since we train to full convergence in both stages, comparing the accuracy (i.e., showing how much it drops from 100% after stage 1) reveals the impact of stage 2 training. We agree that the trend may not be monotonic for individual datasets, but we found that the general trend holds for factoid vs non-factoid datasets.
>
>
> **Increased training budget in REMIX**
>
> We investigated the effect of different mixing ration in Figure 11 (Appendix B.7). While we use a mixing ratio of 2.0 in our main experiments, we found that a mixing ratio of 1.0 can be effective for most settings except S1: Random Mixing / S2: No Mixing, indicating a promising direction to further reduce the mixing ratio if cost is of major concern.
> In addition, since the setting we consider is in the low data regime (2000 examples per stage), the extra training cost is not significant.
>
> **Overlap in the mixing data**
>
> Thank you for pointing out this potential source of confusion. The entity overlap ratio calculated in Appendix B.10 is an overestimate. For example, parsing the instance “Question: Behind Russia, what is the second largest country in Europe? Answer: Ukraine” results in entities like “Russia”, “Europe”, and “Ukraine”. It’s easy to find in generic corpora a text that mentions all the entities, yet the relation “the second largest" is not present. This is different from the replay data which is the exact instance that contains this relation.If we apply strict entity parsing rules and only extract tags such as PERSON, PRODUCT, ORG, then the overlap is close to zero (0.1%).
>
> We have updated them in the revision.
>
> **Typos in section 2.1 and section 4.1 (page 5)**
>
> Thanks for pointing out the errors. We have updated them in the revision.

---

### Review · Reviewer_HoCV · 2025-07-27

**Summary Of Contributions:**

This paper studies forgetting in "continual memorization" setting. As such, the authors study why this forgetting happens (due to task interference) and propose methods called "REMIX" to mitigate this forgetting. To that extent, they propose mixing pertaining data/ random data in every stage of training that acts as "anchor" and prevents catastrophic forgetting. Through clever use of "Logit lens" they show that using REMIX diversifies the knowledge in the entire network, rather than concentrating it later layers, which is more susceptible to forgetting when fine tuning on knowledge intensive tasks.

**Audience:**

Yes

**Broader Impact Concerns:**

No ethical concerns.

**Claims And Evidence:**

Yes

**Requested Changes:**

While this maybe out of scope of current paper, it would be good to study the effectiveness of this approach in a general purpose LLM, not in a small scale Q/A type of setting.

**Strengths And Weaknesses:**

Strengths:
1) Very well motivated problem statement and clear evidence of why this problem exists (in Introduction section)
2) Solid suite of experiments to show why existing methods (replay, regualrization) are not effective in continual memorization problem
3) Well motivated REMIX solution to mitigate forgetting in continual learning setting
4) Use of Logit lens is a great tool to analyze the effectiveness of the method
5) The Ideas presented in the paper are general purposed and can be applicable to other fields like safety tuning and instruction fine-tuning.

Weakness:
* While the claims and the study are inspiring, the experimental setup is very small-scale. The authors study the effectiveness of their approach in a very small scale setting of Q/A type tasks. An LLM is trained and more importantly fine-tuned for range of capabilities  including Q/A, reasoning, translation, coding etc. It would be good to study this problem in a general purpose LLM where they are pre-trained on a combination of tasks, and fine-tuned on a combination of tasks. For example, can we apply these methods in safety tuning of LLMs?

---

> ### Author Response · Authors · 2025-11-21
> **Response to Reviewer HoCV**
>
> We thank the reviewer for recognizing the strengths of our paper including the importance of the problem and the soundness of our experiments and analysis.
>
> **General setting beyond QA**
>
> The key ways our work differs from prior continual learning approaches is that we identify the unique challenge of continually learning factual knowledge. In Table 1, we show that forgetting is generally more severe when learning QA data continually (i.e., factoid data), and that training on regular instruction tuning has lesser impact on the previously learned factoids. We hope this work addresses continual learning on knowledge as a standalone challenge.
> We agree that learning general data beyond factual knowledge is an important direction in continual learning and really appreciate the suggestions. However, we believe that this direction is orthogonal to our main investigation.

---

### Review · Reviewer_Dx18 · 2025-11-09

**Summary Of Contributions:**

This paper addresses the forgetting problem caused by experience replay methods and proposes an effective data mixing strategy to mix random word sequences or generic pretraining data into different training stages. It formalizes the continual memorization setting and demonstrates the fragility of the factoid memorization process in language models. The experimental results show that the proposed method outperforms replay-based methods.

**Audience:**

Yes

**Claims And Evidence:**

Yes

**Requested Changes:**

1. Figure 2 only shows the accuracy changes across different replay ratios but does not compare with the accuracy changes in data mixing, thus, it’s hard to see how replay is worse than mixing. It should compare these two methods under four mixing ratios.


2. On page 5, “The mixing data is sampled from either random word sequences or generic text such as pretraining corpora, which has no overlap with the factoids aiming to memorize in stage 1.” Can we view these mixing data as factoids in stage 0? Since it can be from pretraining corpora, which is earlier than stage 1. If so, does it mean we can mix factoid data from stage $i-1$ with stage $i+1$?



3. When explaining the intuition behind the mixing strategies on page 5, (1) is “$\mathcal{L}(\theta_B;D_A) -\mathcal{L}(\theta_B;D_A)$” correct expression in “the increase in loss after stage 2 is $\mathcal{L}(\theta_B;D_A) -\mathcal{L}(\theta_B;D_A)$? (2) The authors use gradient descent to analyze, but in most cases, the model uses stochastic gradient descent to update. Can the analysis in GD be expanded to SGD?


4. On page 6, “We use DA : DM = 1 : 2 and DB : DM = 1 : 2 for the main experiments.” How to obtain this ratio? Does this ratio depend on specific datasets? Is there any theoretical analysis to explain this ratio?

**Strengths And Weaknesses:**

Strengths:

1. The paper distinguishes mixing and replay to propose an empirical data mixing method to retain previous knowledge and learn new knowledge.

2. The paper is well written and easy to follow.

Weaknesses:

1. The paper lacks a theoretical analysis of why and how data mixing can improve the performance in continual learning.

2. The experiments lack other continual learning method baselines, making it hard to see how REMIX performs compared to other non-replay-based methods.

---

> ### Author Response · Authors · 2025-11-21
> **Response to Reviewer Dx18**
>
> We thank the reviewer for the detailed comments and suggestions. We provide the response to the individual points below.
>
> **Theoretical justification of the approach / can the analysis in GD be expanded to SGD?**
>
> We outline our approach in section 4.1, and have provided the theoretical derivation and analysis in Appendix A.3. Since the main evidence provided in the paper remains empirical, we defer the theoretical analysis in the appendix as a supplementary for establishing intuition for the proposed method. We have organized the section and formulated the descriptions into a proposition in the revision to make it clearer.
>
> We use GD in our theoretical analysis for the purpose of simplifying the derivations in Appendix A.3 following practices in prior theory work [1, 2]. In practice, we use mini-batch gradient updates throughout our experiments.
>
> [1] A Kernel-Based View of Language Model Fine-Tuning by Malladi et al.
> [2] A Qualitative Study of the Dynamic Behavior for Adaptive Gradient Algorithms by Ma et al.
>
>
> **Comparison to other continual learning baselines**
>
> We have considered three representative continual learning baselines in Table 4. In particular, we include (1) weight based regularization (Kirkpatrick et al. (2017)), (2) behavior regularization (Sun et al. (2020)), and parameter expansion methods (von Oswald et al., 2020). Our results show that REMIX outperforms weight based regularization and behavior regularization by large margins, and has a smaller yet notable benefit over the parameter expansion baseline.
>
> **Comparison to replay methods**
> >Figure 2 only shows the accuracy changes across different replay ratios but does not compare with the accuracy changes in data mixing, thus, it’s hard to see how replay is worse than mixing. It should compare these two methods under four mixing ratios.
>
> We would like to clarify that we do not assume access to factoids in the earlier stage in the main settings and experiments (Table 2 and Table 4).
> In the continual memorization setting, the factoids are disjoint instances to memorize and the goal is to achieve perfect retention. In this setting, replay can be considered a form of “cheating”---i.e., training on the exact instances that we are evaluating retention on. For example, replay at 20% means a guaranteed 20% accuracy after stage 2, which is an unfair advantage over the other methods. This is a distinction between the continual memorization and the typical continual learning setting.
> This is why we only use replay as a motivating setting (up to 10% mixing) in Section 3.2 to highlight the challenge as opposed to a major point of comparison.
>
> **Mixing factoids from different stages**
> > Can we view these mixing data as factoids in stage 0? If so, does it mean we can mix factoid data from stage i - 1 with i + 1?
>
> In our setting, the factoids learned in stage 1 ($D_A$) should be kept away from later stage training because we aim to evaluate the retention accuracy on $D_A$. If $D_A$ is mixed into the later stages, then the evaluation becomes “leaky”. While some mixing data may contain factual information (e.g., K-pile mixing) and can be thought of as stage 0 factoids conceptually, we do not consider mixing factoids in stage 1 ($D_A$) into later stages.
>
>
> **Determining mixing ratio**
> > On page 6, “We use DA : DM = 1 : 2 and DB : DM = 1 : 2 for the main experiments.” How to obtain this ratio? Does this ratio depend on specific datasets? Is there any theoretical analysis to explain this ratio?
>
> We decide these ratios based on their empirical effectiveness. We provide the mixing ratio sweep in Figure 11 (Appendix B.7). We found that in some cases, a higher mixture ratio (>2) can be more effective for stage 1, and a lower mixture ratio (<2) can be more effective for stage 2. Based on the sweep, we found that 2 is a reasonable middle ground to demonstrate the effectiveness and therefore fix this hyperparameter throughout.
>
> **Typo in page 5**
> Thank you for pointing out the error. We have updated the error in the revision.

---

### Decision · Action_Editor_HNWQ · 2026-02-10

**Recommendation:** Accept with minor revision

**Additional Comments:**

Congratulations! The reviewers all agreed that this paper is a good contribution to TMLR.

**Audience:**

Yes

**Audience Explanation:**

The reviewers were unanimous that this paper, in the field of continual learning would be of interest to that community.

**Claims And Evidence:**

Yes

**Claims Explanation:**

The reviewers mostly agreed that the claims are well supported.

Some questions remain about the validity of the conclusions drawn from the results in Table 1 (see the reviewer's S9M8 initial review).

In short, I would ask the authors to update Section 3.1, which discusses the results in Table 1. Here are the points that I think are worth keeping in mind:
+ The bias in this evaluation. D_b's are all different, so this is not an "apples to apples" comparison.
+ The individual inconsistencies. The authore recognize this in their response.
+ Consider adding some variability estimate next to the reported average and recognizing that this is more of an initial empirical signal than a fact.